# Huntingtin structure is orchestrated by HAP40 and shows a polyglutamine expansion-specific interaction with exon 1

Rachel J. Harding [1✉], Justin C. Deme [2,3,4], Johannes F. Hevler [5,6], Sem Tamara [5,6], Alexander Lemak[7], Jeffrey P. Cantle[8], Magdalena M. Szewczyk[1], Nola Begeja[9], Siobhan Goss[9], Xiaobing Zuo [10], Peter Loppnau[1], Alma Seitova[1], Ashley Hutchinson[1], Lixin Fan[11], Ray Truant[9], Matthieu Schapira [1,12], Jeffrey B. Carroll[8], Albert J. R. Heck [5,6], Susan M. Lea [2,3,4] & Cheryl H. Arrowsmith [1,7✉]

Huntington's disease results from expansion of a glutamine-coding CAG tract in the huntingtin (HTT) gene, producing an aberrantly functioning form of HTT. Both wildtype and disease-state HTT form a hetero-dimer with HAP40 of unknown functional relevance. We demonstrate in vivo and in cell models that HTT and HAP40 cellular abundance are coupled. Integrating data from a 2.6 Å cryo-electron microscopy structure, cross-linking mass spectrometry, small-angle X-ray scattering, and modeling, we provide a near-atomic-level view of HTT, its molecular interaction surfaces and compacted domain architecture, orchestrated by HAP40. Native mass spectrometry reveals a remarkably stable hetero-dimer, potentially explaining the cellular inter-dependence of HTT and HAP40. The exon 1 region of HTT is dynamic but shows greater conformational variety in the polyglutamine expanded mutant than wildtype exon 1. Our data provide a foundation for future functional and drug discovery studies targeting Huntington's disease and illuminate the structural consequences of HTT polyglutamine expansion.

[1] Structural Genomics Consortium, University of Toronto, Toronto, ON M5G 1L7, Canada. [2] Sir William Dunn School of Pathology, University of Oxford, South Parks Road, Oxford OX1 3RE, UK. [3] Central Oxford Structural Molecular Imaging Centre, University of Oxford, South Parks Road, Oxford OX1 3RE, UK. [4] Center for Structural Biology, Center for Cancer Research, National Cancer Institute, Frederick, MD 21702, USA. [5] Biomolecular Mass Spectrometry and Proteomics, Bijvoet Center for Biomolecular Research and Utrecht Institute of Pharmaceutical Sciences, Utrecht University, Padualaan 8, 3584 CH Utrecht, The Netherlands. [6] Netherlands Proteomics Center, Padualaan 8, 3584 CH Utrecht, The Netherlands. [7] Princess Margaret Cancer Centre and Department of Medical Biophysics, University of Toronto, Toronto, ON M5G 1L7, Canada. [8] Behavioral Neuroscience Program, Department of Psychology, Western Washington University, Bellingham, WA 98225, USA. [9] Department of Biochemistry and Biomedical Sciences, McMaster University, Hamilton, ON L8S 4K1, Canada. [10] X-ray Science Division, Argonne National Laboratory, Lemont, IL 60439, USA. [11] Basic Science Program, Frederick National Laboratory for Cancer Research, SAXS Core of NCI, National Institutes of Health, Frederick, MD 21701, USA. [12] Department of Pharmacology and Toxicology, University of Toronto, Toronto, ON M5S 1A8, Canada. ✉email: Rachel.Harding@utoronto.ca; Cheryl.Arrowsmith@uhnresearch.ca

The autosomal-dominant neurodegenerative disorder Huntington's disease (HD) is caused by the expansion of a CAG repeat tract at the 5′ of the *huntingtin* gene above a critical threshold of ~35 repeats[1]. CAG tract expansion corresponds to an expanded polyglutamine tract of the Huntingtin (HTT) protein, which functions aberrantly compared to its unexpanded form[2]. Polyglutamine expanded HTT is thought to be responsible for disrupting a wide range of cellular processes, including proteostasis[3,4], transcription[5,6], mitochondrial function[7], axonal transport[8] and synaptic function[9]. HD patients experience a range of physical, cognitive and psychological symptoms and longer repeat expansions are associated with earlier disease onset[10]. The prognosis for HD patients is poor, with an average life expectancy of just 18 years from the point of symptom onset. There are currently no disease-modifying therapies available to HD patients.

HTT is a 3144 amino acid protein comprised of namesake HEAT (Huntingtin, Elongation factor 3, protein phosphatase 2A, TOR1) repeats and is hypothesised to function as a scaffold for larger multi-protein assemblies[11,12]. Many proteomics and interaction studies suggest HTT has an extensive interactome of hundreds of proteins but the only biophysically and structurally validated interactor of HTT is the so-called 40-kDa HTT-associated protein HAP40[13,14], an interaction partner conserved through evolution[15,16]. HAP40 is a TPR domain protein with suggested functions in endocytosis[17–19]. An earlier 4 Å mid-resolution cryo-electron microscopy (cryo-EM) model of HTT in complex with HAP40 reveals that the HEAT subdomains of HTT wrap around HAP40 across a large interaction interface[20]. Biophysical and biochemical analyses comparing purified HTT and HTT-HAP40 samples have revealed that HAP40-bound forms of HTT exhibit reduced aggregation propensity, greater structural stability and monodispersity as well as conformational homogeneity[20,21]. Consequently, apo HTT is a more difficult sample to work with for structural and biophysical characterisation, and several studies to date have required cross-linking approaches to constrain the HTT molecule to facilitate its analysis, suggesting HTT-HAP40 interactions may stabilise HTT[22,23]. The biological function of the HTT-HAP40 complex however, remains elusive, and it is not clear if the function of this complex differs from apo HTT in vivo. It is also not yet understood whether HTT is constitutively bound to HAP40 or whether apo and HAP40-bound forms of HTT perform different functions in the cell.

Current structural information for the full-length HTT molecule provide little information on the N-terminal exon 1 region of the protein spanning residues 1–90, which contains the critical polyglutamine and polyproline tracts. This region of the protein is unresolved in the HTT-HAP40 cryo-EM model (PDBID: 6EZ8)[20] and therefore the influence of the tract expansion on HTT structure–function remains the subject of investigation. Although many studies have focussed on understanding the effects of polyglutamine expansion on exon 1 in isolation[24–26], there is still very little known about exon 1 in the context of the full-length HTT protein molecule, either in the apo form or in the complex with HAP40. The intrinsically disordered region (IDR), which spans residues 400–660 is subject to a range of post-translational modifications (PTMs), is postulated to be critical in mediating various protein interactions[21,27,28], and is also unresolved in the cryo-EM structure. Understanding the function of both wild-type (WT) and expanded forms of HTT is critical as many potential HD treatments currently under clinical investigation aim to lower HTT expression, using both allele selective or non-selective approaches[29]. Deeper biological insight into the determinants of cellular HTT protein levels, as well as normal and expanded HTT cellular function would help direct which approaches should be prioritised for long-term patient therapies.

Here we report in vivo studies that show a strong correlation of HTT and HAP40 levels in different genetic backgrounds, providing evidence for the importance of the HTT-HAP40 complex in a physiological setting. Combining the power of multiple complementary structural techniques, we have built a model of the missing regions of our high-resolution (2.6 Å) model of HTT-HAP40, including the biologically critical exon 1 region of HTT and the N-terminal region of HAP40. We demonstrate the remarkable stability of the HTT-HAP40 complex, potentially explaining in vivo codependence of these two proteins and providing important insight for future drug developments in pursuit of treating HD.

## Results

**HAP40 levels are dependent on HTT.** The HTT-associated protein HAP40 co-evolved with HTT[15] and a HAP40 orthologue has been identified in many species, including invertebrates[16]. This suggests that HTT and HAP40 may have functions and/or physical interactions that are co-dependent. To investigate the in vivo relationship between HTT and HAP40, we analysed the levels of both proteins and their mRNA transcripts in liver tissue from mice with biallelic knock out of *Htt* compared to that of WT mice (Fig. 1a–d). Hap40 protein levels were significantly reduced in hepatocyte-specific *Htt* knock out mice compared to WT mice, and a statistically significant correlation was observed between the levels of Htt and Hap40 protein levels. Importantly, the mRNA transcript levels of Hap40 did not change appreciably in the knockout mice, and the Htt and Hap40 transcript levels are not correlated.

Next, we assessed changes to HTT and HAP40 protein levels in hTERT-immortalised RPE1 cells following treatment with the HTT-lowering drug branaplam. Branaplam is a splicing modulator that lowers expression of the HTT protein by changing pre-mRNA transcript processing to retain a poison exon leading to the inclusion of a premature STOP codon[30]. HAP40 is an intron-less gene coded entirely within intron 22 of the F8A gene so is not anticipated to be a target of branaplam[19]. Following treatment, HTT levels were significantly lowered in a dose-dependent manner, even at very low doses of branaplam. HAP40 levels, as measured by western blot analysis, were also significantly lower in an apparent dose-dependent manner and a statistically significant correlation was observed between HTT and HAP40 levels as a function of dose (Fig. 1e, f). Together, these data suggest that HAP40 protein stability and/or abundance is dependent on HTT protein levels.

**High-resolution structure of HTT-HAP40 complex.** HTT-HAP40 was expressed in insect cells and purified as previously described[21]. We determined the structure of HTT-HAP40 (PDBID: 6X9O) to a nominal resolution of 2.6 Å using cryo-EM (Fig. 2a, b and Supplementary Fig. 1), improving substantially upon the previously published 4 Å model (PDBID: 6EZ8)[20] and two recent models (PDBIDs: 7DXJ [3.6 Å] and 7DKK [4.1 Å])[31]. Like all previous models, flexible regions accounting for ~25% of the HTT-HAP40 complex, including exon 1 and the IDR, were not resolved in our high-resolution maps (Fig. 2c). However, our improved resolution permits more confident positioning of amino acid side chains in the structured regions thereby enabling more precise analysis of key structural features and surfaces.

The overall structure of the complex is similar to the previously published model (PDBID: 6EZ8) with a root-mean-square deviation of 1.9 across the models when superposed. However,

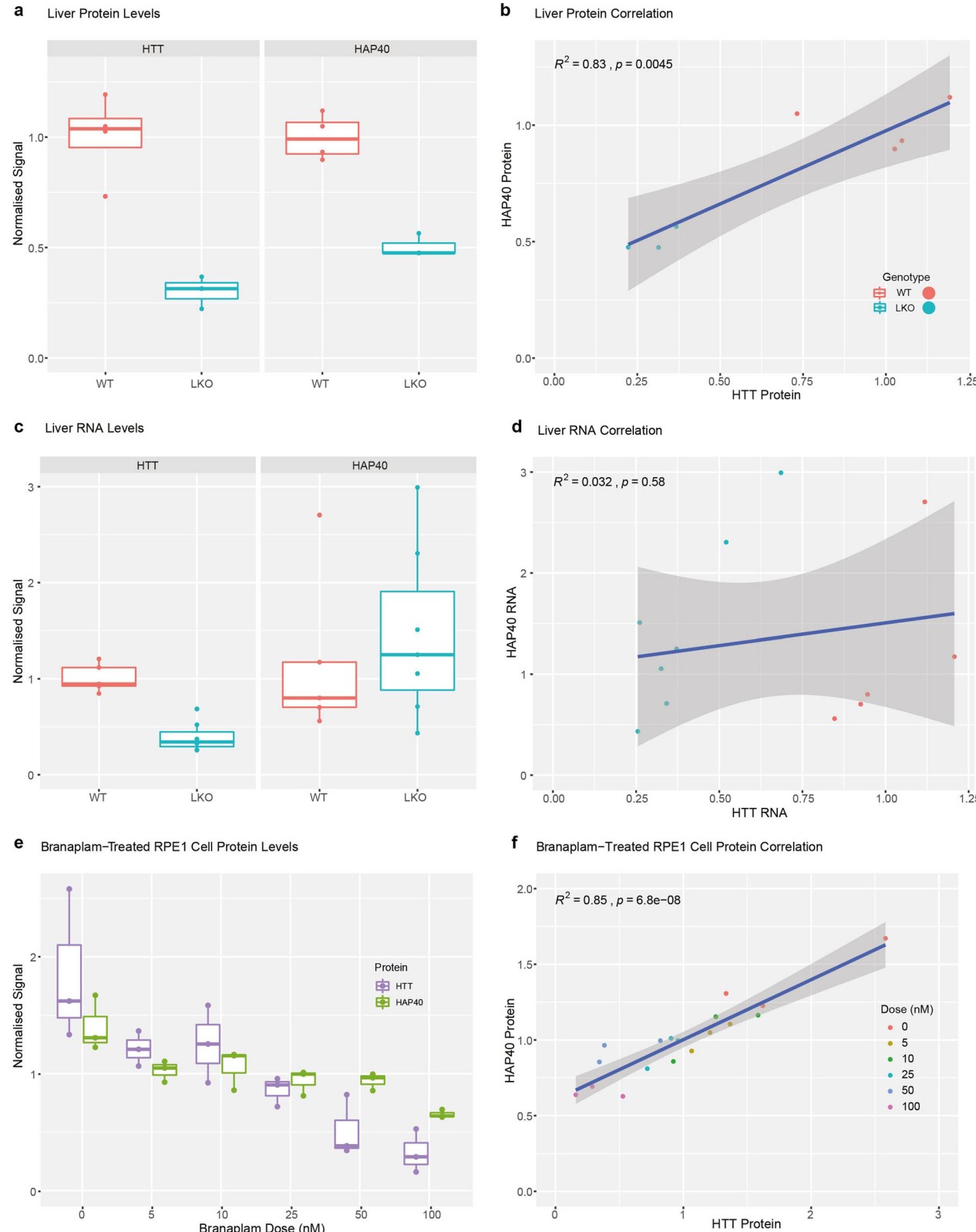

key differences exist between the two models (Fig. 2d). Two additional C-terminal α-helices in the HTT C-HEAT domain spanning residues 3105–3137 are resolved in our model (all residue numbering based on HTT NCBI reference NP_002102.4 sequence), whereas the resolution of two N-terminal α-helices of HAP40 spanning residues 42–82 is lost. The unmodified native HAP40 C-terminus in our model is able to

thread into the centre of the C-HEAT domain (Fig. 2e). This extended interaction of HAP40 with HTT may be responsible for a small shift we observe of the C-HEAT domain, which pivots ~5° relative to the previous model, reducing the interaction interface of HTT-HAP40 from ~5350 to ~4700 Å$^2$. One potential reason for this difference is that the C-terminus of HAP40 in our construct is unmodified, whereas Guo and colleagues[20] used a

**Fig. 1 HTT lowering reduces HAP40 protein levels but not mRNA levels in vivo. a** Htt and Hap40 protein levels in the liver of WT and hepatocyte-specific Htt knockout (LKO) mouse. Hepatocytes make up roughly 80% of liver mass[82], and LKO liver tissue shows ~70% reduction in HTT levels while HAP40 is reduced ~50%. The reduction in both instances is statistically significant (unpaired $t$ test; $t_{(4.0407)} = 6.6184$, $p = 0.002607$ and $t_{(4.5721)} = 8.3375$, $p = 0.0006203$, respectively). **b** HTT lowering drives reduction of HAP40 protein ($R^2 = 0.83$, $F_{(1,5)} = 24.05$, $p = 0.0045$). **c** Liver Htt and Hap40 RNA levels assessed by quantitative PCR. Htt RNA levels are reduced ~60% in LKO mice (unpaired $t$ test; $t_{(9.039)} = 6.9096$, $p = 6.845 \times 10^{-05}$), while Hap40 RNA levels remain unchanged. **d** Lowering Htt RNA levels does not drive Hap40 RNA reduction ($R^2 = 0.03$, $F_{(1,10)} = 0.347$, $p = 0.5757$). **$p < 0.01$, ***$p < 0.001$. **e** Western blot quantified HTT and HAP40 protein levels in human RPE1 cells after 72-h treatment with the HTT-lowering drug branaplam or DMSO control (0 nM). **f** Data points from **e** plotted showing HTT lowering by branaplam drives reduction of HAP40 protein ($R^2 = 0.85$, $F_{(1,16)} = 87,67$, $p = 6.8 \times 10^{-08}$). For **a, c, e**, boxplot hinges correspond to the first and third quartiles of the data, with the horizontal line indicating the median. Whiskers correspond to the smallest and largest datapoint within 1.5 times the interquartile range from the corresponding hinge. Datapoints laying outside the whiskers are >1.5 times the interquartile range from the corresponding hinge.

C-terminal Strep-tag in their expression construct, which is unresolved in their model. The differences observed for the HTT and HAP40 interface when comparing our high-resolution structural model (PDBID: 6X9O) and the previous mid-resolution model (PDBID: 6EZ8) indicate that the extensive interaction interface is able to accommodate some variation.

Our high-resolution model enables a comprehensive analysis of the surface-charge features of the HTT-HAP40 complex. The HTT–HAP40 interface is predominantly formed by extensive hydrophobic interactions between the two proteins (Fig. 2f). Previous analysis of this interface has also highlighted a charge-based interaction between the BRIDGE domain of HTT and the C-terminal region of the HAP40 TPR domain[20]. Interestingly, the N-HEAT domain of HTT has a defined positively charged tract spanning almost 40 Å in length and 5–10 Å in width formed between two stacked HEAT repeats in the N-HEAT solenoid (Fig. 2f arrow). We also conducted sequence conservation analysis of both HTT and HAP40, which we mapped to the high-resolution structure of the complex (Supplementary Data 1 and 2). Interestingly this revealed surfaces on the HAP40-exposed face of the protein as highly conserved, with extended regions of strict conservation partially spanning the C-HEAT domain, BRIDGE and N-HEAT (Fig. 2g). However, the opposite face is less conserved, while the HTT–HAP40 interface is moderately conserved for both HTT and HAP40. The HTT-HAP40 model was analysed for ligand-able pockets, which were assessed for druggability according to factors such as the volume and depth of apparent pockets and their surface charge and hydrophobicity properties. One of the most promising pockets, which is predicted to be ligand-able, lies at the HTT-HAP40 interface and is lined by residues from the N-terminal region of the HAP40 TPR domain as well as the HTT N-HEAT domain (Fig. 2h and Supplementary Table 1). The high resolution of our HTT-HAP40 model provides a foundation for virtual screening of such pockets and other structure-based drug-discovery efforts towards the identification of HTT ligands, which may be developed into proteolysis-targeting chimeras[32] or PET ligands[33] to function as, or monitor, HTT-lowering therapies[29], a critical area of focus for HD drug discovery.

Our 2.6 Å structure is of sufficient resolution to allow the identification of PTMs. However, no PTMs were observed for any of the resolved residues in the HTT-HAP40 complex. Native mass spectrometry (MS) analysis, on the other hand, revealed the high purity of our HTT-HAP40 samples, albeit that a small mass difference (compared to the theoretical mass) was observed, consistent with the presence of a few PTMs (Supplementary Fig. 2a, d). Further analysis of the HTT-HAP40 complex upon Caspase6 digestion revealed these PTMs to be primarily phosphorylations (at least two), which could be mapped to the regions spanning 586–2647 and 2647–3144 of the HTT sequence (Supplementary Fig. 2b–d). Based on the cumulative evidence from the MS data, these modifications reside within the two

flexible portions of HTT not resolved in our cryo-EM maps. Although many studies have identified numerous different sites and possible PTMs of the HTT protein[21,27,28,34], these approaches have so far been qualitative and do not give us a good understanding of the key proteoforms the HD community is studying in either in vitro or in vivo models. Our quantitative top–down and middle–down MS approaches suggest many PTMs are in fact only present at very low abundance, at least in our samples expressed in insect cells.

We attempted to separately purify HTT and HAP40 for comparison to the complex. As reported by Guo and colleagues[20], we were also unable to express recombinant HAP40 alone, although it is readily expressed in the presence of HTT, a trend that parallels our in vivo observations and cell biology experiments. In the absence of HAP40, we and others have shown that recombinant HTT self-associates and is conformationally heterogenous in vitro[21,22,34]. Cryo-EM analysis of our apo HTT samples yielded a 12 Å resolution envelope (Fig. 3a, b and Supplementary Data 3). Despite the low resolution of this envelope, it is possible to identify the N-HEAT domain, with its central cavity, as well as the C-HEAT domain. The HTT portion of our HTT-HAP40 model can be fitted into this envelope. Comparison of this envelope with the previously reported apo HTT cryo-EM envelopes that were stabilised by cross-linking (EMD4937 and EMD10793)[22] shows a less collapsed arrangement of the HTT subdomains. The difference in resolution between apo HTT and HTT-HAP40 samples observed by cryo-EM analysis emphasises the importance of HAP40 in stabilising the HTT protein and constraining the HEAT repeat subdomains into a more rigid conformation, further supporting the idea that this is a critical interaction for modulating HTT structure and function.

Native top–down MS uses gas-phase activation to dissociate protein complexes enabling identification of complex composition and subunit stoichiometries. The most commonly used activation method using collisions with neutral gas molecules typically results in dissociation of a non-covalent complex into constituent subunits. Interestingly, our native top–down MS analysis of the intact HTT-HAP40 complex (Fig. 4a, b) primarily resulted in backbone fragmentation of HTT, eliminating both N- and C-terminal fragments (Fig. 4c–g). Remarkably, the vast majority of concomitantly formed high-mass dissociation products retained HAP40 (Fig. 4f), suggesting that the extensive hydrophobic interaction interface we observe in our high-resolution model keeps the HTT-HAP40 complex exceptionally stable. Similarly, gas-phase activation of Caspase6-treated HTT-HAP40 revealed that HAP40 remained intact and bound to HTT even at the highest activation energies, whereas the N- and C-terminal fragments of HTT produced upon digestion were readily dissociating from the complex (Supplementary Fig. 2c).

The recombinant samples of HTT-HAP40 were found to be highly monodisperse (Fig. 4b), displaying optimal biophysical

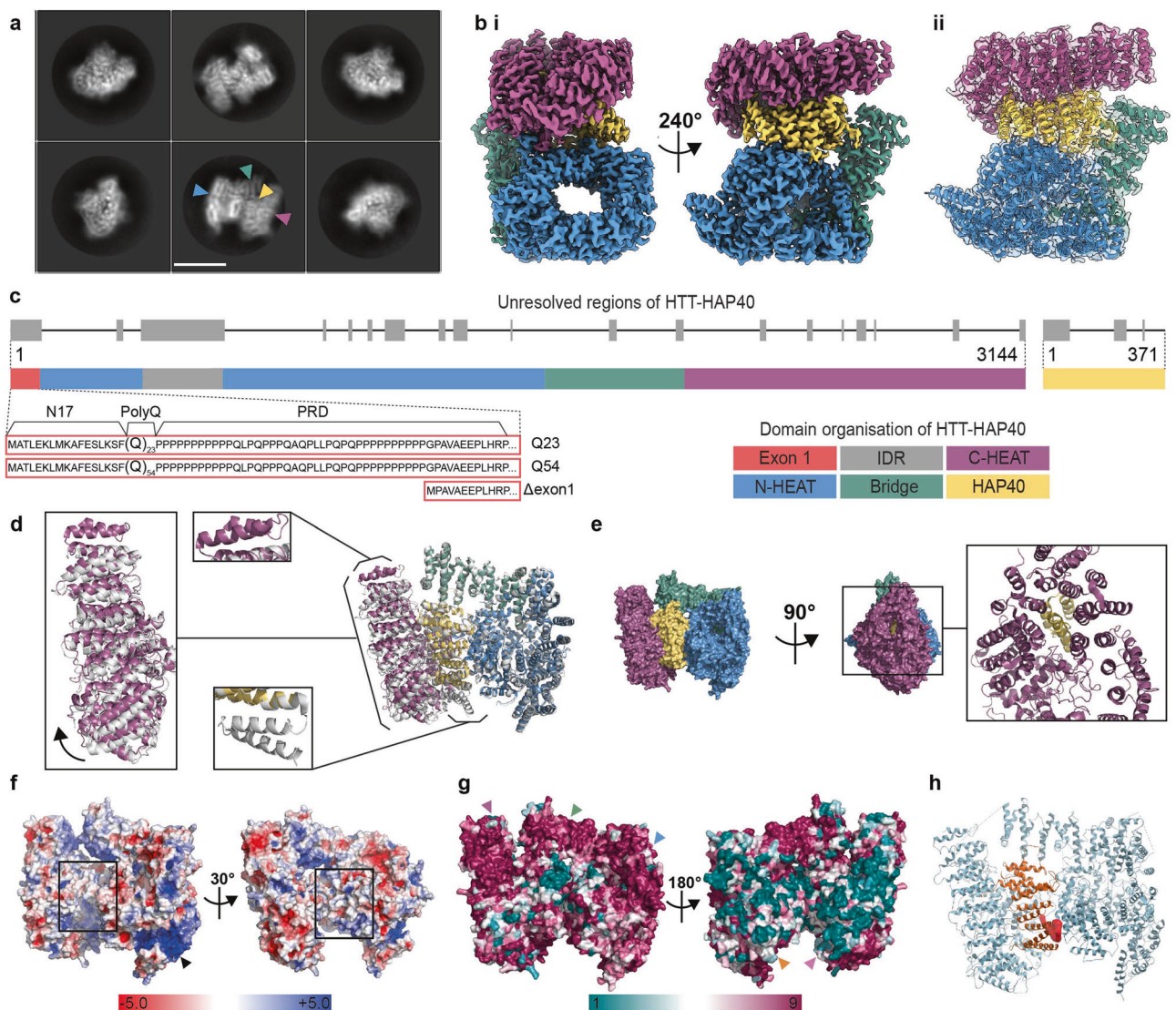

**Fig. 2 HAP40 stabilises the structure of HTT via extensive interactions across all subdomains. a** Representative cryo-EM 2D class averages of HTT-HAP40. Scale bar (white) is 90 Å. Blue and purple arrowheads denote N- and C-HEAT domains of HTT, respectively. Green and yellow arrowheads denote bridge domain of HTT and HAP40, respectively. **b** Cryo-EM volume of HTT-HAP40 resolved to 2.6 Å with (i) HTT N-HEAT in blue, bridge domain in green, C-HEAT in purple and HAP40 in yellow or (ii) map shown with HTT-HAP40 modelled in using the same domain colour convention. **c** Domain organisation of HTT shown mapped to linear sequence. Unresolved regions of the structure are in grey and the three different constructs used in this study are detailed comprising wild-type (23 glutamines; Q23), mutant (54 glutamines; Q54), or HTT with exon 1 partially deleted (Δexon 1; comprising residues 80–3144). **d** Superposition of our model (PDBID: 6X9O—same domain colour convention as before) and the previous model (PDBID: 6EZ8—all grey) with alignment calculated over N-HEAT and bridge domains. Additional α-helices observed in either of the models are indicated with boxes, C-HEAT domain shift is shown with an arrow. **e** Surface representation of HTT and HAP40 (same domain colour convention as before) in front and side views, rotated 90°, with additional panel (right) showing same side view of the complex in cartoon format. **f** Electrostatic surface representation of HTT with HAP40 removed from the structure. Positively charged regions are shown in blue, neutral regions in white and negatively charged regions in red. The positively charged tract in the N-HEAT domain is indicated with a black arrowhead. Hydrophobic HTT surface that binds HAP40 is indicated with hollow black boxes. **g** Surface representation of HTT-HAP40 complex, coloured according to Consurf conservation scores: from teal for the least conserved residues (1), to maroon for the most conserved residues (9). Conserved surfaces for C-HEAT, bridge and N-HEAT domains are indicated with purple, green and blue arrowheads, respectively. Variable N-HEAT and C-HEAT surfaces are indicated with orange and pink arrowheads, respectively. **h** HTT (pale blue)-HAP40 (orange) complex in cartoon with pocket predicted to be druggable shown as red volume.

properties (see also Supplementary Fig. 3a). Systematically screening the thermal stability of the HTT-HAP40 complex using a differential scanning fluorimetry (DSF) assay indicates that the complex is highly stable under a broad range of buffer, pH and salt conditions (Supplementary Fig. 3b, c). Destabilisation of the complex structure was only observed at low pH (Fig. 4h). Similarly, the interaction between HTT and HAP40 is retained upon mild proteolysis of the complex (Fig. 4I, j, all data in

Supplementary Fig. 3d). For example, in an attempt to fragment HTT with Caspase6 treatment as previously described[35], we found that the HTT-HAP40 complex remains associated under native conditions. The same samples, when analysed under denaturing conditions used in western blots, showed apparent HTT cleavage products. These observations suggest caution when drawing conclusions about proteolytic fragments of HTT observed in western blot analyses of biological samples. Taken

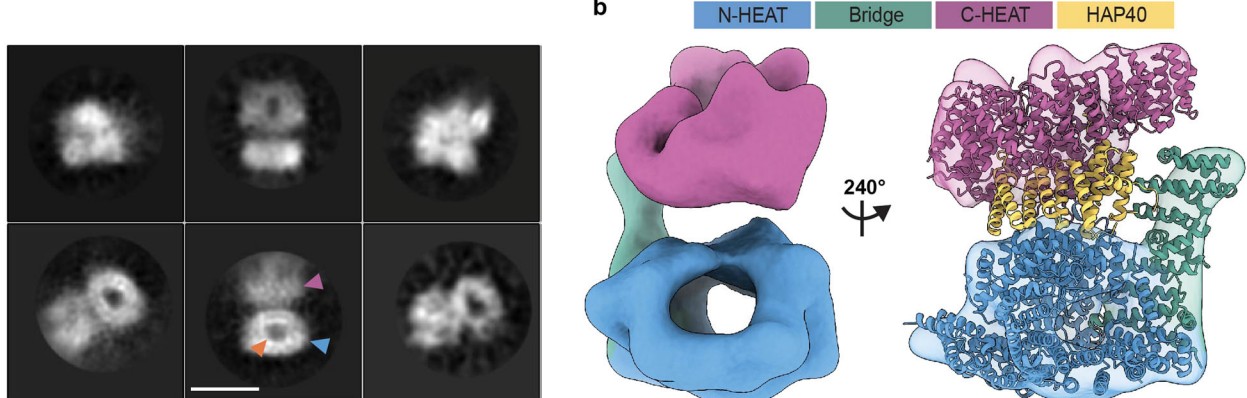

**Fig. 3 HTT HEAT domains are conformationally flexible in the absence of HAP40. a** Representative cryo-EM 2D class averages of HTT Q23. Scale bar (white) shown in the bottom middle panel is 90 Å. Blue arrowhead denotes the N-HEAT domain in which its central cavity (orange arrowhead) is more clearly defined. Purple arrowhead denotes the less well-defined C-HEAT domain, perhaps due to conformational flexibility relative to the N-HEAT. **b** Cryo-EM volume of HTT resolved to ~12 Å shown with model of HTT-HAP40 (PDBID: 6X9O) fit to the map. Regions of map and model are displayed with N-HEAT in blue, bridge domain in green, C-HEAT in purple and HAP40 in yellow.

together, our studies reveal the high structural stability of the HTT-HAP40 complex with resistance to dissociation by native top–down MS, or proteolytic cleavage in solution. These data further support the high codependence of HTT and HAP40 protein levels in cell and animal models of HD and possibly HD patients.

**Polyglutamine expansion modulates the dynamic sampling of conformational space by exon 1.** Next, we sought to understand how the disease-causing polyglutamine expansions affect HTT structure. Our structural, biophysical and biochemical data presented so far focus on WT HTT (23 glutamines; Q23) and illustrate the importance of HAP40 in stabilising and orienting the HEAT repeat subdomains of HTT. However, 25% of the complex is not resolved in the cryo-EM maps, including many functionally important regions of the protein such as exon 1 (residues 1–90), which harbours the polyglutamine repeat region, and the IDR (residues 407–665). To further investigate the HTT protein structure in its entirety and the influence of polyglutamine expansion within exon 1, we repeated the DSF and proteolysis studies using HTT-HAP40 samples containing either a pathological HD HTT with 54 glutamines (Q54) or an HTT with a partially deleted exon 1 (Δexon 1; comprising residues 80–3144, missing N17, polyglutamine and proline-rich domain). We found that both the Q54 expansion and the removal of exon 1 had no detectable effects on the stability of the HTT-HAP40 complexes compared to the canonical Q23 complex (Supplementary Fig. 3).

To better describe the structure of exon 1 and the effects of the polyglutamine expansion on the HTT-HAP40 complex, we performed cross-linking mass spectrometry (XL-MS) experiments[36–38] using the IMAC-enrichable lysine cross-linker, PhoX[39]. For cross-linking experiments, an optimised PhoX concentration was used, for which no cross-linker-induced protein aggregation was observed by mass photometry (Supplementary Fig. 4a). For Q23, Q54 and Δexon 1 isoforms of HTT-HAP40, we mapped approximately 120 cross-links for each sample (Supplementary Data File 7) which were highly reproducible (Supplementary Fig. 4b, c). When analysed against the 6X9O model of HTT-HAP40, the vast majority of cross-links map to regions unresolved in the cryo-EM maps (Fig. 5a), thereby providing valuable restraints for structural modelling of a more complete HTT-HAP40 complex. The mean distance of cross-links observed for resolved regions of the cryo-EM model was

significantly below the 25 Å distance limit of PhoX in all three data sets (Q23: 7 cross-links—mean distance 13.7 Å; Q54: 11 cross-links—mean distance 14.8 Å; Δexon 1: 12 cross-links—mean distance 14.9 Å; Supplementary Data File 7). This is in line with the mass photometry data and indicates that there is a low probability of intermolecular cross-links between HTT molecules, e.g. from aggregation, being included in our data sets (Supplementary Fig. 4a).

We obtained very similar cross-link data for the three different HTT-HAP40 constructs (Supplementary Fig. 5b), which span all subdomains of HTT and also HAP40, indicating good cross-linking efficiency (Supplementary Fig. 5a, c). Of particular note are the large number of exon 1 PhoX cross-links in the HTT-HAP40 Q23 and Q54 samples mediated via lysine-6 or lysine-9 within the N-terminal 17 residues (N17 region) of exon 1 (Fig. 5b). N17 is reported to play key roles for the HTT protein including modulating cellular localisation, aggregation and toxicity[40–42] and is proposed to interact with distal parts of HTT[43].

For both samples (Q23 and Q54), N17 is found to contact several regions of the N-HEAT domain as well as the cryo-EM unresolved N-terminal region of HAP40, via lysine-32 and lysine-40. Interestingly, N17 of Q54 showed additional cross-links to the more distant C-HEAT domain (Fig. 5b and Supplementary Fig. 4b). Finally, the largest uninterrupted stretch of the HTT-HAP40 protein that is unresolved in the cryo-EM maps is the IDR. We identified cross-links which indicate that this region makes intra-domain contacts as well as contacts with the neighbouring N-HEAT domain.

Size-exclusion chromatography multi-angle light scattering (SEC-MALS) analysis of this same series of samples shows no significant difference in mass but does indicate a small shift in the peak for the elution volume of the HTT-HAP40 Δexon 1 complex compared to Q23 and Q54 complex samples (Fig. 6a). Together with the XL-MS data, this suggests that there are subtle structural differences between the Q23, Q54 and Δexon 1 HTT-HAP40 complexes. To further interpret the cross-linking data in the context of the three-dimensional (3D) structure of the HTT-HAP40 complex, we performed small-angle X-ray scattering (SAXS) analysis of our samples to assess any changes to their global structures (Supplementary Data 4–6). We have previously reported SAXS data for HTT-HAP40 Q23[21]. This revealed that the particle size was significantly larger than the cryo-EM model, which likely accounts for the ~25% of the protein not resolved in

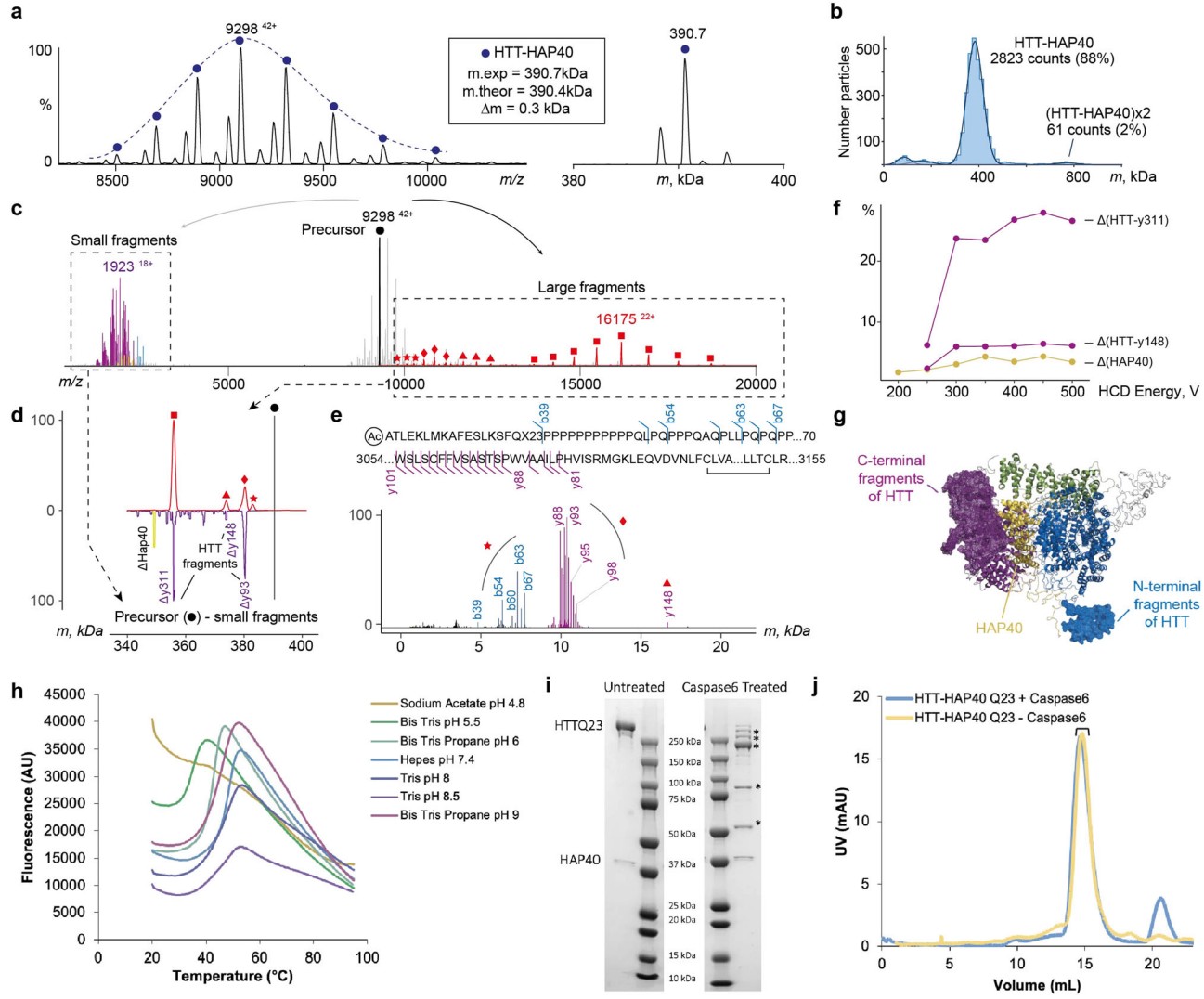

**Fig. 4 HTT and HAP40 form a very stable non-covalent complex that withstands dissociation. a** Raw native (left) and a deconvoluted zero-charged (right) spectrum of the HTT-HAP40 Q23 complex. **b** Mass profile of HTT-HAP40 complex obtained using mass photometry, showing that the complex is monodisperse. **c** Composite native top–down mass spectrum of the HTT-HAP40 complex demonstrating large (right of the precursor) and small (left of the precursor) dissociation products produced at the highest activation energy. The data reveal that N- and C-terminal fragments of HTT are eliminated from the HTT-HAP40 complex upon collisional activation, whereas the intact HAP40 remains bound. Small fragment peaks are coloured following domain colour convention for the HTT-HAP40 complex. **d** Upper mass profile represents deconvoluted masses (red) of the large dissociation products (**c** in red) experimentally obtained by activating HTT-HAP40 complexes in the gas phase using HCD. Mirrored fragment masses (purple) are obtained by subtracting masses of the small experimental fragments (**c** in purple) from the precursor mass. **e** Annotation of small fragments obtained at high-resolution settings and mapping to the sequence of HTT Q23. Red stars, diamonds, triangles, and squares in panels (**c–e**) denote distinct dissociation products of HTT-HAP40 and link them from **c** (raw peaks) to d (deconvoluted masses) to **e** (complementary low-mass fragments). **f** Energy-resolved plot of fragment abundances: HTT with HAP40 ejected (yellow), HTT upon release of C-terminal fragment y311 or y148 (purple). **g** Structure of HTT-HAP40 complex with eliminated regions highlighted and represented as mesh. Colour-coding is in accordance with the domain colour convention for HTT-HAP40. **h** Assessing HTT-HAP40 Q23 complex stability by measuring transition temperature using DSF in different buffer conditions with 300 mM NaCl. **i** Caspase6 digestion of HTT-HAP40 Q23 proteins assessed by SDS-PAGE and **j** analytical gel filtration. Peak fractions from gel filtration run on SDS-PAGE are indicated and SDS-PAGE identified cleavage products are indicated (*).

cryo-EM maps and therefore not modelled in the structure. Similar analysis of the HTT-HAP40 Q54 and HTT-HAP40 Δexon 1 and comparison with our previous Q23 data shows that polyglutamine expansion or deletion of exon 1 has only very modest effects on the SAXS profiles (Fig. 6b–d). HTT-HAP40 Q54 is slightly larger than the HTT-HAP40 Q23, whereas HTT-HAP40 Δexon 1 samples are slightly smaller, as might be expected, but overall the SAXS-determined parameters for the three samples are very similar (Fig. 6e) as we would expect for samples with highly similar structural cores[31]. In line with that,

the SAXS-calculated particle envelopes for the three samples are also very similar in size and shape (Supplementary Fig. 6a).

Next, we modelled the complete structures of HTT-HAP40, including flexible and disordered regions, integrating our cryo-EM, SAXS and XL-MS data, similar to other studies using integrated approaches to study disordered protein structures[37,38]. Coarse-grain modelling molecular dynamics (MD) simulations were performed and an ensemble of models that best fit both the cross-linking and SAXS data for HTT-HAP40 was calculated for all three variants of the HTT-HAP40 complex (Supplementary

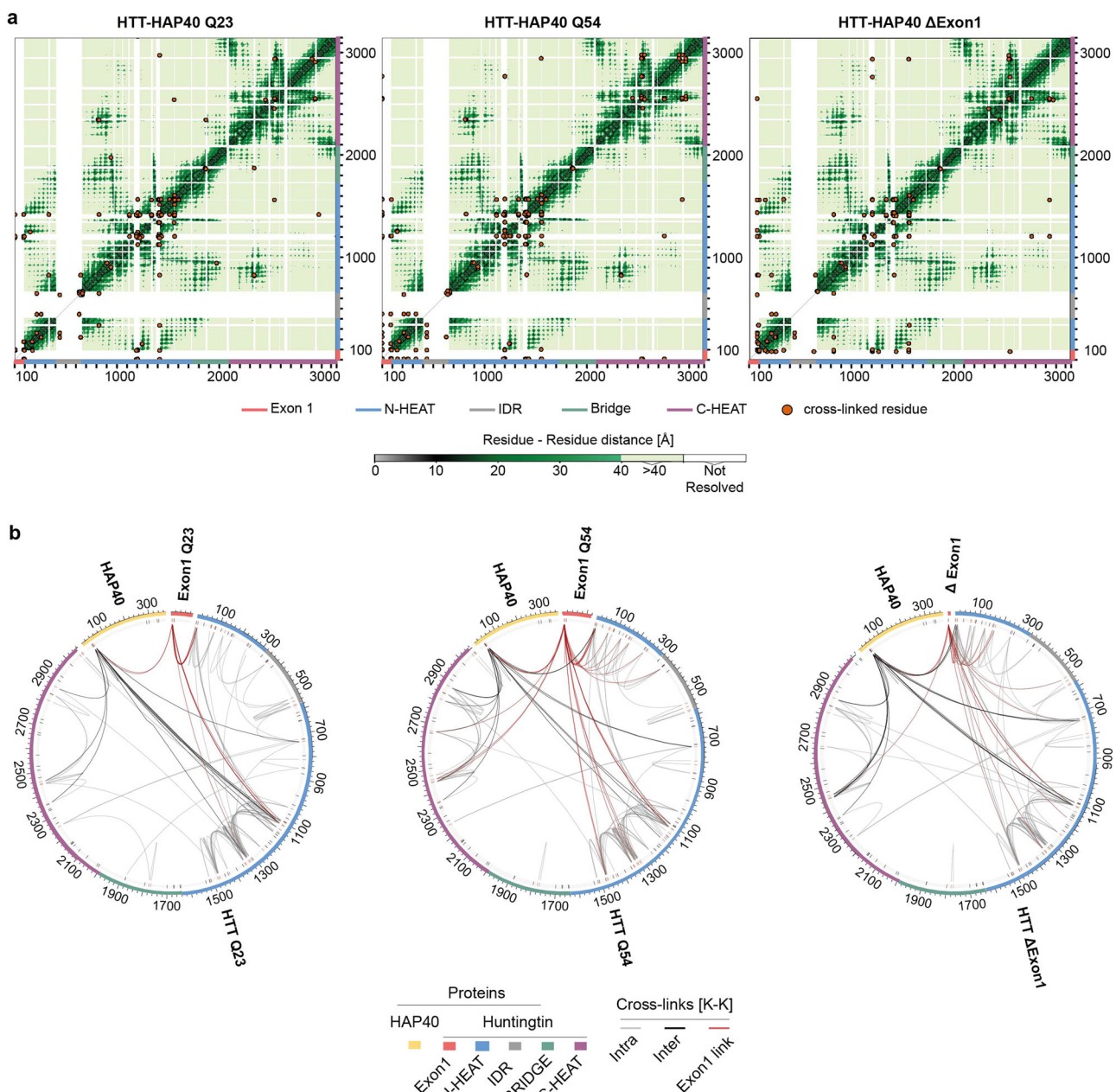

**Fig. 5 Exon 1 is highly flexible and conformationally dynamic in the context of the full-length protein. a** Mapping cross-linked sites to the HTT-HAP40 sequence of different samples, with cross-linked residue pairs shown as orange circles. Intramolecular distances for HTT-HAP40 (PDBID: 6X9O) shown from grey to green as per the coloured scale bar with unmodelled regions of the protein shown in white. **b** Mapping cross-links to the HTT-HAP40 sequence of different samples, with exon 1 in red, N-HEAT in blue, bridge domain in green, IDR in grey, C-HEAT in purple and HAP40 in yellow. Cross-linked lysine residues are indicated in red and unmodified lysine residues are indicated in black on the numbered sequence. Intermolecular cross-links (HTT-HAP40) are shown in black, intramolecular cross-links (HAP40-HAP40 or HTT-HTT) are shown in grey and exon 1 cross-links are shown in red. All residues following the exon 1 region of the different constructs are numbered the same for clarity.

Fig. 6b, c and Supplementary Data 8–11). This modelling approach assumed that the residues with known coordinates in the cryo-EM model form a quasi-rigid complex, whereas the residues with missing coordinates are flexible. As expected from our cross-linking results, the suggested conformations adopted by exon 1 in the ensemble model of Q54 HTT-HAP40 complex are skewed compared to the Q23 ensemble with exon 1 interacting with many more surfaces of the Q54 HTT-HAP40 complex (Fig. 7a). Mapping our PhoX exon 1 cross-linked residues for each sample to a representative model from each ensemble reveals how exon 1 Q23 cross-links are largely constrained to the

N-HEAT domain, whereas exon 1 Q54 cross-links are also found on the C-HEAT domain (Supplementary Fig. 4b). Exon 1 of our HTT-HAP40 Q54 ensemble appears to explore a larger volume of conformational space, and this seems to have a knock-on effect on the conformational space occupied by the IDR (Fig. 7b).

Modelling of our HTT-HAP40 structure indicates that the exon 1 region of the Q23 HTT is long enough to make cross-links with the C-HEAT domain, but we do not observe such cross-links in our PhoX data sets. This suggests that the additional cross-links observed for the polyglutamine expanded form of HTT-HAP40 may not be driven solely by the length of the exon 1

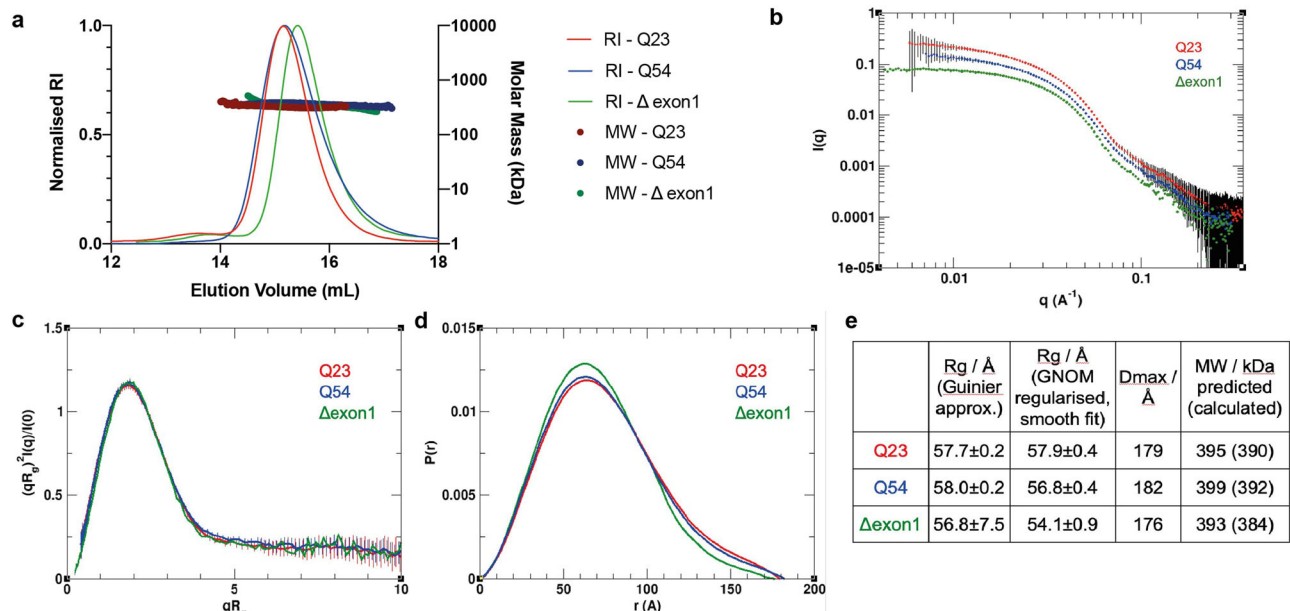

**Fig. 6 Polyglutamine expansion or deletion of exon 1 has modest effects on the full-length HTT-HAP40 SAXS profile. a** SEC-MALS analysis of HTT-HAP40 samples Q23 (red), Q54 (blue) and Δexon 1 (green). **b** Experimental SAXS data. **c** $R_g$-based (dimensionless) Kratky plots of experimental SAXS data for HTT-HAP40 Q23 (red), Q54 (blue) and Δexon 1 (green). **d** Normalised pair distance distribution function $P(r)$ calculated from experimental SAXS data with GNOM for HTT-HAP40 samples. **e** SAXS parameters for data validation and interpretation including radius of gyration ($R_g$) calculated using Guinier fit in the $q$ range $0.015 < q < 0.025$ Å$^{-1}$, radius of gyration calculated using GNOM, maximum distance between atoms calculated using GNOM and the molecular mass estimated using SAXSMoW with expected masses from the respective construct sequences shown in the parentheses.

region. For all ensembles, the IDR is differentially constrained and occluded from adopting certain conformations depending on the conformational space occupied by exon 1, suggesting that polyglutamine and exon 1-mediated structural changes propagate to the IDR. Despite exon 1 and the IDR being separated by the first HEAT repeat domain spanning aa. 98–406, our modelling suggests that they are proximal in the full-length HTT-HAP40 complex, indicating that structural changes to one of these regions has the potential for a knock-on effect on the other. For the HTT-HAP40 Q54 model ensemble which suggests exon 1 adopts the most diverse conformations, the IDR is the most constrained, occupying a more finite space. However, for the HTT-HAP40 Δexon 1 model ensemble, the IDR is not occluded and so adopts a much wider range of conformations.

We used unconstrained MD simulations to analyse the contact frequency of experimentally identified cross-links in exon 1 and the IDR of HTT-HAP40 Q23 and Q54 (Supplementary Tables 2–5). All cross-links experimentally identified for the exon 1 region of HTT-HAP40 Q23 are also observed for the Q54 form of the protein. However, exon 1–C-HEAT cross-links that are uniquely identified in our HTT-HAP40 Q54 experiments have similar frequencies in our unconstrained simulations of HTT-HAP40 Q23 and Q54, supporting our conclusion that the experimental identification of exon 1–C-HEAT cross-links for HTT-HAP40 Q54 is significant (Supplementary Fig. 7a). Similarly, analysis of experimentally identified IDR cross-links in these simulations shows that the unique exon 1–IDR cross-link in the HTT-HAP40 Q54 data set has a similar frequency in both simulations, again indicating that experimental identification of this cross-link shows a significant difference in the conformational space occupied by exon 1 upon polyglutamine expansion (Supplementary Fig. 7b). These findings also support our use of experimental constraints, including SAXS and cross-linking data, to generate more realistic and statistically representative ensembles of the possible conformations of all flexible regions of HTT-

HAP40 to help identify the subtle structural differences caused by polyglutamine expansion.

Together, our data suggest that, while polyglutamine expansion does not affect the core HEAT repeat structure, it does affect the conformational space occupied by not only the exon 1 region but also the IDR.

## Discussion
We present new insights for the HTT-HAP40 structure, highlighting the close relationship between HTT and HAP40 as well as unveiling the effect of the polyglutamine expansion, thereby contributing to a richer understanding of HTT and its relationship with HAP40.

HTT is reported to interact with hundreds of different proteins[14] but very few have been validated and the only interaction partner resolved by structural methods is HAP40. HAP40 is thought to have coevolved with HTT[15] and orthologues have been identified in species back to flies[16]. The codependence of HTT and HAP40 is highlighted with our in vivo analysis of HTT and HAP40 levels in mice, which shows a strong correlation of the two proteins. We also demonstrate that this relationship is independent of the RNA transcript levels of the HTT and HAP40. Additionally, we showed in human cells that pharmacological lowering of HTT protein levels with the drug branaplam also reduced HAP40, and a statistically significant correlation ($R^2 = 0.85$) was observed between HTT and HAP40 levels as a function of dose. Overall, this suggests that HAP40 protein stability and/or abundance is dependent on HTT protein levels.

It remains to be seen whether HTT and HAP40 are in fact constitutively bound to each other, or if they may exist independently or in complex with other binding partners. HAP40 plays an important role in stabilising HTT conformation as we have shown with our biophysical and structural comparison of apo and HAP40-bound HTT samples, but the molecular mechanisms of how HAP40 functions in endosome transport[17,18]

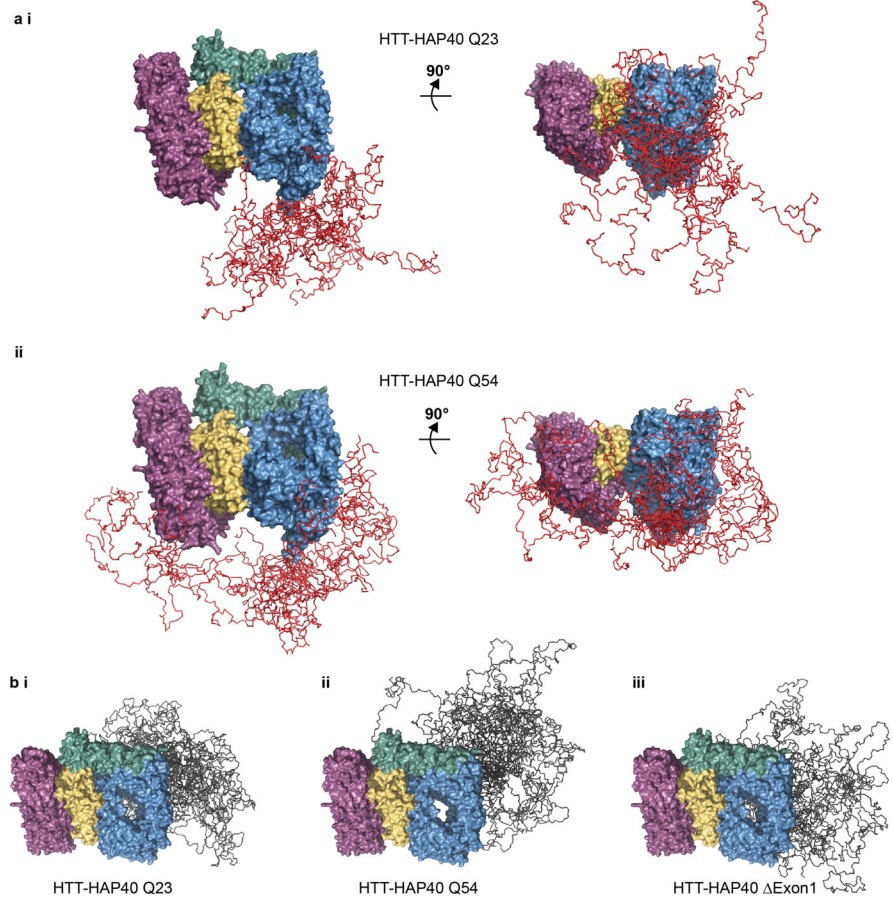

**Fig. 7 Insights from integrated modelling of full-length HTT-HAP40 combining cryo-EM, SAXS and cross-linking mass spectrometry data. a** Ensemble of models for HTT-HAP40 (i) Q23 and (ii) Q54 showing only the residues defined by the cryo-EM model in surface representation (N-HEAT in blue, bridge domain in green, C-HEAT in purple and HAP40 in yellow) and exon 1-simulated residues in ribbon representation (red). **b** Ensemble of models for HTT-HAP40 (i) Q23, (ii) Q54 and (iii) Δexon 1 showing only the residues defined by the cryo-EM model in surface representation (N-HEAT in blue, bridge domain in green, C-HEAT in purple and HAP40 in yellow) and IDR-simulated residues in ribbon representation (grey).

or modulating HTT toxicity in HD models[16] remains to be determined. Interestingly, despite the exceptional stability of the HTT–HAP40 interaction, complex integrity was not maintained in our DSF assay at low pH, conditions similar to that of the local environment of the endosome.

How polyglutamine expansion of HTT contributes to changes in protein structure–function remains a critical and unanswered question in HD research. Previously, we have observed that changes in polyglutamine tract length seem to have minimal effects on the biophysical properties of HTT and HTT-HAP40 samples[21] and studies have shown that these regions are dispensable for HTT function in mice[44]. Similarly, in this study, we find no significant differences between our Q23, Q54 and Δexon 1 HTT-HAP40 samples when assessing monodispersity by mass photometry and native MS: thermal stability in a systematic buffer screen by DSF or stabilisation by proteolysis experiments. The structural differences of Q23, Q54 and Δexon 1 HTT-HAP40 samples are not resolved within the high-resolution cryo-EM maps we calculated. Our experiments using lower-resolution structural methods such as SAXS and mass spectrometry, which do consider the complete protein molecule, also show modest differences between the samples. One way we might rationalise this observation with what we know about HD pathology and HTT biology in physiological conditions is that our experimental systems do not capture any subtle, low abundance or slowly occurring differences of the samples which could be important in HD progression that occurs very slowly, over decades of a

patient's lifetime. Alternatively, it may be that models of HD pathogenesis which posit that large changes in HTT's globular structure caused by polyglutamine expansion[23] are incorrect.

Notwithstanding the above caveats, our XL-MS studies provide some of the first insight into the structure of the exon 1 portion of the protein in the context of the full-length, HAP40-bound form of HTT. In both Q23 and Q54 samples, exon 1 appears to be highly dynamic and able to adopt multiple conformations. Our data suggest differences in the conformational ensembles of the unexpanded and expanded forms of exon 1 in the context of the full-length HTT protein (Supplementary Movie 1). Specifically, the expanded Q54 forms of exon 1 appear to sample different conformational space than unexpanded Q23. This is not just due to the additional length of this form of exon 1, conferring a higher degree of flexibility and extension to different regions of the protein but perhaps some biophysical consequence of a longer polyglutamine tract. This is the opposite of what has been reported for HTT exon 1 protein in isolation, where polyglutamine expansion compacts the exon 1 structure[45–48]. Our data suggest that in the context of the full-length HAP40-bound HTT protein, exon 1 is not compact, but flexible and conformationally dynamic while retaining moderate structural organisation. Our modelling studies interestingly suggest that the change in exon 1 conformational sampling upon polyglutamine expansion may have consequent effects on the relative conformations and orientations of the IDR, a novel insight to HTT structure. Both exon 1 and the IDR have been highlighted as

functionally important regions of HTT, as sites of dynamic PTMs and protease recognition concentrate in these regions. Our results suggest that structural changes in exon 1 induced by poly-glutamine expansion could influence the accessibility of the IDR to partner proteins that modify residues within the IDR, despite the relatively rigid intervening regions between them. The flex-ibility we observe for exon 1 in both Q23 WT and Q54 mutant HTT-HAP40 supports the hypothesis that polyglutamine tracts can function as sensors, sampling and responding to their local environment[49].

Overall, our findings show that HTT is stabilised by interaction with HAP40 through an extensive hydrophobic interface with its distinct HEAT repeat subdomains, creating a highly stable complex. Expanded and unexpanded exon 1 remains highly dynamic in the context of this complex, sampling a vast range of conformational space and interacting with different regions of both HTT and HAP40. We present novel insight into the struc-tural differences of WT and mutant HTT, which suggests that the conformational constraints of WT and mutant exon 1 are dif-ferent and that models of HD pathogenesis relying on the hypothesis that polyglutamine expansion drives large-scale changes in HTT conformation may need to be re-examined.

## Methods

**In vivo HTT-HAP40 protein and RNA transcript levels.** To generate samples with genetic reduction of HTT levels in the liver, mice in which the first exon of *Htt* is flanked by LoxP sites[50] were crossed with mice expressing CRE recombinase from the *Alb* promoter (JAX:003574). Fresh frozen livers from WT and LKO mice were collected at 5–6 months of age for subsequent protein and RNA analyses. Protein lysates were prepared for western blotting using non-denaturing lysis buffer (20 mM Tris HCl pH 8, 127 mM NaCl, 1% NP-40, 2 mM EDTA), with 50 ug of protein separated using 3–8% tris-acetate gels (Invitrogen EA0378) and trans-ferred using an iBlot2 transfer system (Invitrogen IB21001). Probing with anti-bodies against HTT (Abcam EPR5526; 1:1000) and HAP40 (Novus NBP2-54731; 1:500) was performed overnight at 4 C with gentle shaking, followed by incubation with near infrared secondary antibodies (Licor 926–68073; 1:10,000). Signal was normalised to total protein in the lane (Licor 926–11010). Imaging was performed using an Odyssey imager and signal quantitated using ImageStudio (Licor). RNA samples were prepared using RNeasy Lipid Mini Kit (Qiagen) with on-column DNase digestion. cDNA was prepared using Superscript III (Invitrogen 18080-051) with random hexamer primers. Multiplexed quantitative PCR was performed on a Step One thermocycler (Applied Biosystems) using TaqMan assays and 200 ng cDNA template per reaction. FAM-labelled Htt and Hap40 probes were run in separate reactions, each with VIC-labelled ActB probes as internal control (Ther-moFisher Mm01213820_m1, Mm03016217_s1, and Mm02619580_g1, respec-tively). Each biological replicate represents the average of three technical replicates, with relative quantification performed using delta-delta Ct calculation[51]. Mouse strain: C57Bl/6J. Sex: western blot all females; qPCR mixed male/female. Age: western blot 5 months +/− a week; qPCR 5.5 months +/− 2 weeks. All procedures were reviewed and approved by the animal care and use committee at Western Washington University.

**Cell culture and branaplam treatment experiments.** The hTERT-immortalised retinal pigment epithelial cell line, RPE1 (ATCC—CRL-4000) was cultured in Dulbecco's Modified Eagle Medium/F12 1:1 media supplemented with 10% fetal bovine serum and 0.01 mg/mL hygromycin B at 37 °C in a 5% CO$_2$ incubator. Cell lines were not authenticated after purchase from supplier. Cells were negative for mycoplasma. At approximately 40% cell confluency, cells were treated with bra-naplam (Selleckchem—(6E)-3-(1H-Pyrazol-4-yl)-6-[3-(2,2,6,6-tetra-methylpiperidin-4-yl)oxy-1H-pyridazin-6-ylidene]cyclohexa-2,4-dien-1-one) dis-solved in dimethyl sulfoxide and then diluted with culture media to final concentrations of 5, 10, 25, 50, and 100 nM for 72 h. Cells were prepared for western blotting using RIPA buffer (50 mM Tris-HCl pH 8.0, 150 mM NaCl, 1% NP-40, 0.25% sodium deoxycholate, 1 mM EDTA, protease and phosphatase inhibitors (Roche) with 60 μg protein separated by sodium dodecyl sulfate-polyacrylamide gel electrophoresis (SDS-PAGE) on a 4–20% gradient gel (Bio-Rad) and transferred to a polyvinylidene difluoride membrane (Millipore). Membranes were blocked in TBS-T (50 mM Tris-HCl, pH 7.5, 150 mM NaCl, 0.1% Tween-20) containing 5% skim milk powder for 1 h then cut into three sections to be probed with primary antibodies against HTT (Millipore MAB2166; 1:2500), HAP40 (LS-C167891, LSBio; 1:500), and vinculin (EPR8185, Abcam; 1:2500) in the same buffer overnight at 4 °C. Membranes were washed four times with TBS-T and then probed with horseradish peroxidase secondary antibodies (Abcam) for 30 min at room temperature (RT). After washing as above, membranes were incubated with ECL (Millipore) and imaged with a DNR MicroChemi chemiluminescence

detector. Signal was quantified with ImageJ using the "Analyze Gel" option. HTT and HAP40 signals were normalised to the vinculin signal from the corresponding lane. Experiments were completed with three independent replicates.

**Protein expression constructs.** HTT Q23, HTT Q54 and HAP40 constructs used in this study have been previously described[21] and are available through Addgene with accession numbers 111726, 111727 and 124060, respectively. HTT Δexon 1 clones spanning HTT aa. 80–3144 were also cloned into the pBMDEL vector. A PCR product encoding HTT from residues P76 to C3140 was amplified from cDNA (Kazusa clone FHC15881) using primers FWD (ttaagaaggagata-tactatgCCGGCTGTGGCTGAGGAGC) and REV (gattggaagta-gaggttctctgcGCAGGTGGTGACCTTGTGG). PCR products were inserted using the In-Fusion cloning kit (Clontech) into the pBMDEL that had been linearised with BfuAI. The HTT-coding sequences of expression constructs were confirmed by DNA sequencing. The sequences were also confirmed by Addgene where these reagents have been deposited. This clone is available through Addgene with accession number 162274.

**Protein expression and purification.** HTT and HTT-HAP40 protein samples were expressed in insect cells and purified using a similar protocol as previously described[21]. Briefly, Sf9 cells were infected with P3 recombinant baculovirus and grown until viability dropped to 80–85%, normally after ~72 h post-infection. For HTT-HAP40 complex production, a 1:1 ratio of HTT:HAP40 P3 recombinant baculovirus was used for infection. Cells were harvested, lysed with freeze–thaw cycles and then clarified by centrifugation. HTT protein samples were purified by FLAG-affinity chromatography. FLAG eluted samples were bound to Heparin FF cartridge (GE) and washed with 10 CV 20 mM HEPES pH 7.4, 50 mM KCl, 1 mM TCEP, 2.5% glycerol and eluted with a gradient from 50 mM KCl buffer to 1 M KCl buffer over 10 CV. All samples were purified with a final gel filtration step, using a Superose6 10/300 column in 20 mM HEPES pH 7.4, 300 mM NaCl, 1 mM TCEP, 2.5% (v/v) glycerol. HTT-HAP40 samples were further purified with an additional Ni-affinity chromatography step prior to gel filtration. Fractions of the peaks corresponding to the HTT monomer or HTT-HAP40 heterodimer were pooled, concentrated, aliquoted and flash frozen prior to use in downstream experiments. Sample purity was assessed by SDS-PAGE. The sample identities were confirmed by native mass spectrometry (Fig. 5).

**SDS-PAGE and western blot analysis.** SDS-PAGE and western blot analysis were performed according to standard protocols. Primary antibodies used in western blots are anti-HTT EPR5526 (Abcam), anti-HTT D7F7 (Cell Signaling Technol-ogies) and anti-Flag #F4799 (Sigma). Secondary antibodies used in western blots are goat-anti-rabbit IgG-IR800 (LI-COR) and donkey anti-mouse IgG-IR680 (LI-COR). Membranes were visualised on an Odyssey® CLx Imaging System (LI-COR).

**DSF analysis of HTT samples.** HTT samples were diluted in different buffer conditions and incubated at RT for 15 min before the addition of Sypro Orange (Invitrogen) to a final concentration of 5×. The final protein concentration was 0.15 mg/mL. Measurements were performed using a Light Cycler 480 II instrument from Roche Applied Science over the course of 20–95 °C. Temperature scan curves were fitted to a Boltzmann sigmoid function, and the transition temperature values were obtained from the midpoint of the transition.

**Caspase6 proteolysis of HTT protein samples.** HTT protein samples were mixed with recombinant Caspase6 (Enzo Life Sciences) in a ratio of 100 U caspase6 to 1 pmol of HTT in 20 mM HEPES pH 7.4, 150 mM NaCl and 1 mM TCEP with a final protein concentration of ~1 μM. The reaction and control mixture without caspase6 were incubated at RT for 16 h and then analysed by SDS-PAGE, blue native PAGE and analytical gel filtration using a Superose6 10/300 column in 20 mM HEPES pH 7.4, 150 mM NaCl and 1 mM TCEP.

**Cross-linking of HTT-HAP40 samples with PhoX.** For cross-linking experi-ments, HTT-HAP40 samples (HTTQ23-HAP40, HTTQ54-HAP40, HTT Δexon 1-HAP40) were diluted to a protein concentration of 1 mg/1 mL using cross-linking buffer (20 mM Hepes pH 7.4, 300 mM NaCl, 2.5% glycerol, 1 mM TCEP). HTT-HAP40 samples were treated with an optimised PhoX cross-linker concentration to avoid protein aggregation (Supplementary Fig. 4a). After incubation with PhoX (0.5 mM) for 30 min at RT, the reaction was quenched for additional 30 min at RT by the addition of Tris HCl (1 M, pH 7.5) to a final concentration of 50 mM. Protein digestion was performed in 100 mM Tris-HCl, pH 8.5, 1% SDC, 5 mM TCEP and 30 mM CAA, with the addition of Lys-C and Trypsin proteases (1:25 and 1:100 ratio (w/w)) overnight at 37 °C. The reaction was stopped by addition of TFA to a final concentration of 0.1% or until pH ~2. Next, peptides were desalted using an Oasis HLB plate, before IMAC enrichment of cross-linked peptides like previously described[39]. Four technical replicates were completed for each form of HTT-HAP40.

**LC-MS analysis of cross-linked HTT-HAP40 samples.** For LC-MS analysis, the samples were re-suspended in 2% formic acid (FA) and analysed using an

UltiMate™ 3000 RSLCnano System (Thermo Fischer Scientific) coupled on-line to either a Q Exactive HF-X (Thermo Fischer Scientific) or an Orbitrap Exploris 480 (Thermo Fischer Scientific). First, peptides were trapped for 5 min in solvent A (0.1% FA in water), using a 100-μm inner diameter 2-cm trap column (packed in-house with ReproSil-Pur C18-AQ, 3 μm) prior to separation on an analytical column (50 cm of length, 75 μM inner diameter; packed in-house with Poroshell 120 EC-C18, 2.7 μm). Peptides were eluted following a 45 or 55 min gradient from 9-35% solvent B (80% ACN, 0.1% FA), or 9-41% solvent B, respectively. On the Q Exactive HF-X, a full-scan MS spectra from 375 to 1600 Da were acquired in the Orbitrap at a resolution of 60,000 with the automatic gain control (AGC) target set to $3 \times 10^6$ and maximum injection time (IT) of 120 ms. For measurements on the Orbitrap Exploris 480, a full-scan MS spectra from 375 to 2200 $m/z$ were acquired in the Orbitrap at a resolution of 60,000 with the AGC target set to $2 \times 10^6$ and maximum IT of 25 ms. Only peptides with charged states 3–8 were fragmented, and dynamic exclusion properties were set to $n = 1$, for a duration of 10 s (Q Exactive HF-X) and 15 s (Orbitrap Exploris 480). Fragmentation was performed using in a stepped HCD collision energy mode (27, 30, 33% Q Exactive HF-X; 20, 28, 36% Orbitrap Exploris 480) in the ion trap and acquired in the Orbitrap at a resolution of 30,000 after accumulating a target value of $1 \times 10^5$ with an isolation window of 1.4 $m/z$ and maximum IT of 54 ms (Q Exactive HF-X) and 55 ms (Orbitrap Exploris 480).

**Data analysis of HTT-HAP40 cross-links**. Raw files for cross-linked HTT-HAP40 samples were analysed using the XlinkX node[52] in the Proteome Discoverer (PD) software suite 2.5 (Thermo Fischer Scientific), with signal to noise threshold set to 1.4. Trypsin was set as a digestion enzyme (max two allowed missed cleavages), the precursor tolerance set to 10 ppm and the maximum false discovery rate set to 1%. Additionally, carbamidomethyl modification (Cystein) was set as fixed modification and acetylation (protein N-terminus) and oxidation (Methionine) were set as dynamic modifications. Cross-links obtained for respective HTTQ-HAP40 samples were filtered (only cross-links identified with an XlinkX score >40 were considered) and further validated using our recently deposited structure of HTTQ23-HAP40 (PDBID: 6X9O) (EMD-22106). Contact maps and circos plots were generated in R (http://www.R-project.org/) using the circlize[53] and XLmaps[54] packages.

**Mass photometry**. Mass photometry analysis was performed on a Refeyn OneMP instrument (Oxford, UK), which was calibrated using a native marker protein mixture (NativeMark Unstained Protein Standard, Thermo Scientific). The marker contained proteins in the wide mass range up to 1.2 MDa. Four proteins were used to generate a standard calibration curve, with following rounded average masses: 66, 146, 480, and 1048 kDa. The experiments were conducted using glass cover-slips, extensively cleaned through several rounds of washing with Milli-Q water and isopropanol. A set of 4–6 gaskets made of clear silicone was placed onto the thoroughly dried glass surface to create wells for sample load. Typically, 1 μL of HTT samples was applied to 19 μL of phosphate-buffered saline (PBS) resulting in a final concentration of ~5 nM. Movies consisting of 6000 final frames were recorded using the AcquireMP software at a 100 Hz framerate. Particle landing events were automatically detected amounting to ~3000 per acquisition. The data were analysed using the DiscoverMP software. Average masses of HTT proteins and HTT-HAP40 complexes were determined by taking the value at the mode of the normal distribution fitted into the histograms of particle masses. Finally, probability density function was calculated and drawn over the histogram to produce the final mass profile. Measurement and analysis of mass photometry data were done for the following samples: HTT-Q23-HAP40, HTT-Q54-HAP40, and HTT-Δexon 1-HAP40.

**Intact mass and middle–down MS sample preparation**. Sample preparation: Samples containing HTT-HAP40 complexes were digested using human Caspase6 (Enzo Life Sciences, Farmingdale, USA) by adding 200 U of the enzyme to the 20 μg of the protein. The mixture was stored in PBS for 24 h. Following the digestion, samples were diluted to the final concentration of 500 ng/μL with 2% FA. Approximately 2 μg of the sample were injected for a single intact mass LC-MS or middle–down LC-MS/MS experiment.

**LC-MS(/MS) for intact and middle–down MS**. Produced peptides of HTT were separated using a Vanquish Flex UHPLC (Thermo Fisher Scientific, Bremen, Germany) coupled on-line to an Orbitrap Fusion Lumos Tribrid mass spectrometer (Thermo Fisher Scientific, San Jose, USA) via reversed-phase analytical column (MAbPac, 1 mm × 150 mm, Thermo Fisher Scientific). The column compartment and preheater were kept at 80 °C during the measurements to ensure efficient unfolding and separation of the analysed peptides. Analytes were separated and measured for 22 min at a flow rate of 150 μL/min. Elution was conducted using A (Milli-Q H2O/0.1% CH2O2) and B (C2H3N/0.1% CH2O2) mobile phases. In the first minute, B was increased from 10 to 30%, followed by 30 to 57% B gradient over 14 min, 1 min 57 to 95% B ramp-up, 95% B for 1 min and equilibration of the column at 10% B for 4 min.

During data acquisition, Lumos Fusion instrument was set to Intact Protein and Low Pressure mode. MS1 resolution of 7500 (determined at 200 $m/z$ and equivalent to 16 ms transient signal length) was used, which enables optimal detection of protein ions above 30 kDa in mass. Mass range of 500–3000 $m/z$, the AGC target of 250%, and a max IT of 50 ms were used for recording of MS1 scans. Two μscans were averaged in the time domain and recorded for the 7500 resolution scans during the LC-MS experiment and 5 μscans when tandem MS (MS/MS) was performed. MS/MS scans were recorded at a resolution setting of 120,000 (determined at 200 m/z and equivalent to 16 ms transient signal length), 10,000% AGC target, 250 ms max IT, and 5 μscans, for the single most abundant peak detected in the preceding MS1 scan. The selected ions were mass-isolated by a quadrupole in a 4 $m/z$ window and accumulated to an estimate of 5e6 ions prior to the gas-phase activation. Two separate LC-MS/MS runs were recorded per sample with either higher-energy collisional dissociation (HCD) or electron transfer dissociation (ETD) used for fragmentation. For ETD, the following parameters were used: ETD reaction time—16 ms, max IT of the ETD reagent—200 ms, and the AGC target of the ETD reagent—1e6. For HCD, 30 V activation energy was used. MS/MS scans were acquired with the minimum intensity of the precursor set to 5e4 and the range of 350–5000 $m/z$ using quadrupole in the high mass isolation mode.

**Data analysis of intact and middle-down MS**. LC-MS data were deconvoluted with ReSpect algorithm in BioPharma Finder 3.2 (Thermo Fisher Scientific, San Jose, USA). ReSpect parameters: precursor $m/z$ tolerance—0.2 Th, target mass—50 kDa, relative abundance threshold—0%, mass range—3–100 kDa; tolerance—30 ppm, charge range—3–100. MS1 and MS2 masses were recalibrated using an external calibrant mixture of intact proteins (PiercePierce™ Intact Protein Standard Mix, Thermo Scientific) measured before and after each HTT sample. Iterative sequence adjustments of putative HTT peptides was done until the exact precursor and fragment masses matched to determine a final set of HTT peptides generated by Caspase6 enzyme. HCD fragments of HTT peptides were used solely to confirm identified sequences. Phosphorylation was matched as 80 Da variable modification mass, added to the mass of the identified HTT peptides. Visualisation was done in R extended with ggplot2 package.

**Native (top–down) MS sample preparation**. Samples were stored at −80 °C in the buffer containing 20 mM HEPES pH 7.4, 300 mM NaCl, 2.5% (v/v) glycerol, 1 mM TCEP. Approximately 40 μg of the HTT-Q23, HTT-Q54, HTT-Δexon 1, and their respective complexes with Hap40 protein were buffer-exchanged into 150 mM aqueous ammonium acetate (pH = 7.5) by using P-6 Bio-Spin gel filtration columns (Bio-rad, Veenendaal, the Netherlands). The protein's resulting concentration was estimated to be ~2–5 μM before native MS analysis. For the recording of denaturing MS, samples were spiked with FA to the final concentration of 2% right before the MS measurement.

**Native (top–down) data acquisition**. HTT-containing samples were directly injected into a Q Exactive Ultra-High Mass Range (UHMR) Orbitrap mass spectrometer (Thermo Fisher Scientific, Bremen, Germany) using in-house pulled and gold-coated borosilicate capillaries. Following mass spectrometer parameters were used: capillary voltage—1.5 kV, positive ion mode, source temperature—250 °C, S-lens radio frequency (RF) level—200, IT—mostly 200 ms, noise level parameter—3.64. In-source trapping with a desolvation voltage of −100 V was used to desolvate the proteinaceous ions efficiently. No additional acceleration voltage was used in the back-end of the instrument. The AGC was switched to fixed. Resolutions of 4375 and 8750 (both at $m/z = 200$ Th) were used, representing 16 and 32 ms transient, respectively. Ion guide optics and voltage gradient throughout the instrument were manually adjusted for optimal transmission and detection of HTT and HTT-HAP40 ions. The HCD cell was filled with Nitrogen, and the trapping gas pressure was set to 3 or 4 setting value, corresponding to ~2e-10–4e-10 mBar for the ultra-high vacuum readout of the instrument. The instrument was calibrated in the $m/z$ range of interest using a concentrated aqueous cesium iodide (CsI) solution. Acquisition of the spectra was usually performed by averaging 100–200 μscans in the time domain. Peaks corresponding to the protein complex of interest were isolated with a 20–Th window for single charge state isolation and a 2000 Th window for charge-state ensemble isolation. In both cases, isolated HTT-HAP40 ions were investigated for dissociation using elevated HCD voltages, with direct eV setting varied in the range 1–500 V. For detection of high-$m/z$ dissociation product ions, mass analyser detection mode and transmission RF settings were set to "high $m/z$". For detection of low-$m/z$ fragment ions, all relevant instrument settings were set to "low $m/z$", and the instrument resolution was increased to 140,000 (at $m/z = 200$ Th).

**Data analysis for native (top-down) MS**. Raw native MS and high-$m/z$ native top–down MS data were processed with UniDec[55] to obtain zero-charged mass spectra. Native top–down MS data recorded with high resolution (140,000) were deconvoluted using the Xtract algorithm within FreeStyle software (1.7SP1; Thermo Fisher Scientific). The resulting zero-charge fragments were matched to the theoretical fragments produced for HTT and HAP40 using in-house scripts with 5 ppm mass tolerance. Theoretical fragment intensities were derived from the corresponding fragments obtained upon deconvolution of raw native mass spectrum. Final visualisation was performed in R extended with ggplot2 library.

**Cryo-EM sample preparation and data acquisition**. HTT was diluted to 0.4 mg/mL in 20 mM HEPES pH 7.5, 300 mM NaCl, 1 mM TCEP and adsorbed to glow-discharged holey carbon-coated grids (Quantifoil 300 mesh, Au R1.2/1.3) for 10 s. Grids were then blotted with filter paper for 2 s at 100% humidity at 4 °C and frozen in liquid ethane using a Vitrobot Mark IV (Thermo Fisher Scientific).

HTT-HAP40 was diluted to 0.2 mg/mL in 25 mM HEPES pH 7.4, 300 mM NaCl, 0.025% w/v CHAPS and 1 mM DTT and adsorbed onto gently glow-discharged suspended monolayer graphene grids (Graphenea) for 60 s. Grids were then blotted with filter paper for 1 s at 100% humidity, 4 °C and frozen in liquid ethane using a Vitrobot Mark IV (Thermo Fisher Scientific).

Data were collected in counting mode on a Titan Krios G3 (FEI) operating at 300 kV with a BioQuantum imaging filter (Gatan) and K2 direct detection camera (Gatan) at ×165,000 magnification, pixel size of 0.822 Å. Movies were collected over 32 fractions at a dose rate of $6.0\,e^-/Å^2/s$, exposure time of 8 s, resulting in a total dose of $48.0\,e^-/Å^2$.

**Cryo-EM data processing**. For apo HTT, patched motion correction and dose weighting were performed using MotionCor implemented in RELION 3.0[56]. Contrast transfer function parameters were estimated using CTFFIND4[57]. Particles were picked in SIMPLE 3.0[58] and processed in RELION 3.0. Six hundred and sixty-nine movies were collected in total and 108,883 particles extracted. Particles were subjected to one round of reference-free two-dimensional (2D) classification against 100 classes ($k = 100$) using a soft circular mask of 180 Å in diameter in RELION. A subset of 25,424 particles were recovered at this stage and subjected to 3D auto-refinement in RELION using a 40 Å low-pass-filtered map of HTT-HAP40 (EMDB 3984) as initial reference. This generated a ~12 Å map based on gold-standard Fourier shell correlation curves using the 0.143 criterion as calculated within RELION.

For HTT-HAP40 (Supplementary Fig. 1), 15,003 movies were processed in real time using the SIMPLE 3.0 pipeline, using SIMPLE-unblur for patched motion correction, SIMPLE-CTFFIND for patched CTF estimation and SIMPLE-picker for particle picking. After initial 2D classification in SIMPLE 3.0 using the cluster2D_stream module ($k = 500$), cleaned particles were imported into RELION and subjected to reference-free 2D classification (k = 200) using a 180 Å soft circular mask. An ab initio map, generated from a selected subset of particles (372,226), was subsequently lowpass filtered to 40 Å and used as reference for coarse-sampled (7.5°) 3D classification ($k = 4$) with a 180 Å soft spherical mask against the same particle subset. Particles (102,729) belonging to the most defined, highest resolution class were selected for 3D auto-refinement against its corresponding map, lowpass filtered to 40 Å, using a soft mask covering the protein which generated a 3.5 Å volume. This map was lowpass filtered to 40 Å and used as initial reference for a multi-step 3D classification ($k = 5$, 15 iterations at 7.5° followed by 5 iterations at 3.75°), with 180 Å soft spherical mask, against the full cleaned data set of 2,240,373 particles. Selected particles (647,468) from the highest resolution class were subjected to masked 3D auto-refinement against its reference map, lowpass filtered to 15 Å, yielding a 3.1 Å volume. CTF refinement using per-particle defocus plus beamtilt estimation further improved map quality to 3.0 Å. Bayesian particle polishing followed by an additional round of CTF refinement with per-particle defocus plus beamtilt estimation on a larger box size (448 × 448) generated a final volume with global resolution of 2.6 Å as assessed by Gold standard Fourier shell correlations using the 0.143 criterion within RELION. Map local resolution estimation was calculated within Relion (Supplementary Fig. 1). Additional rounds of 3D classification using either global/local searches or classification only without alignment did not improve map quality.

**Model building and refinement**. The model for HTT-HAP40 (Table 1) was generated by rigid body fitting the 4 Å HTT-HAP40 model[20] (PDBID: 6EZ8) into our globally-sharpened, local resolution filtered 2.6 Å map followed by multiple rounds of manual real-space refinement using Coot v. 0.95[59] and automated real-space refinement in PHENIX v. 1.18.2–38746[60] using secondary structure, rotamer and Ramachandran restraints. HTT-HAP40 model was validated using MolProbity[61] within PHENIX. Figures were prepared using UCSF ChimeraX v.1.1[62] and PyMOL v.2.4.0 (The PyMOL Molecular Graphics System, v.2.0; Schrödinger).

**SAXS data collection and analysis**. SAXS experiments were performed at beamline 12-ID-B of the Advanced Photon Source (APS) at Argonne National Laboratory. The energy of the X-ray beam was 13.3 keV (wavelength $\lambda = 0.9322$ Å), and two set-ups (small- and wide-angle X-ray scattering) were used simultaneously to cover scattering $q$ ranges of $0.006 < q < 2.6$ Å$^{-1}$, where $q = (4\pi/\lambda)\sin\theta$, and $2\theta$ is the scattering angle. For HTT-HAP40 Q54, 32-dimensional images were recorded for buffer or sample solutions using a flow cell, with an exposure time of 0.8 s to reduce radiation damage and obtain good statistics. The flow cell is made of a cylindrical quartz capillary 1.5 mm in diameter and 10 μm wall thickness. Concentration-series measurements for this sample were carried out at 300 K with concentrations of 0.5, 1.0, and 2.0 mg/mL in 20 mM HEPES, pH 7.5, 300 mM NaCl, 2.5% (v/v) glycerol and 1 mM TCEP. No radiation damage was observed as confirmed by the absence of systematic signal changes in sequentially collected X-ray scattering images. The 2D images were corrected for solid angle of each pixel,

and reduced to one-dimensional (1D) scattering profiles using the Matlab software package at the beamlines. The 1D SAXS profiles were grouped by sample and averaged.

For HTT-HAP40 Δexon 1, data were collected using an in-line FPLC AKTA micro set-up with a Superose6 Increase 10/300 GL size exclusion column in 20 mm HEPES, pH 7.5, 300 mm NaCl, 2.5% (v/v) glycerol and 1 mm TCEP. A 150 μL sample loop was used and the stock sample concentration was 5 mg/mL. The sample passed through the FPLC column and was fed to the flow cell for SAXS measurements. The SAXS data were collected every 2 s and the X-ray exposure time was set to 0.75 s. Only the SAXS data collected above the half maximum of the elution peak, about 50–100 frames, were averaged and used for further analysis. Background data were collected before and after the peak (each 100 frames), while data before the peak were found better and used for the background subtraction.

SAXS data were analysed with the software package ATSAS 2.8[63]. The experimental radius of gyration, $R_g$, was calculated from data at low $q$ values using the Guinier approximation. The pair distance distribution function, $P(r)$, the maximum dimension of the protein, $D_{max}$, and $R_g$ in real space were calculated with the indirect Fourier transform using the program GNOM[64]. Estimation of the molecular weight of samples was obtained by both SAXMOW[65,66] and by using volume of correlation, Vc[67]. The theoretical scattering intensity of the atomic structure model was calculated using FoXS[68]. Ab initio shape reconstructions (molecular envelopes) were performed using both bead modelling with DAMMIF[69] and calculating 3D particle electron densities directly from SAXS data with DENSS[70] (Supplementary Fig. 6a).

**Coarse-grained MD simulations**. We used a Gō-like coarse-grained model of HTT-HAP40 for structural modelling of the complex as described previously[21]. In this model, amino acid residues in the proteins are represented as single beads

**Table 1 Cryo-EM data collection, refinement and validation statistics.**

| | HTT-HAP40 (EMDB-22106) (PDB 6X9O) |
|---|---|
| *Data collection and processing* | |
| Magnification | 165,000 |
| Voltage (kV) | 300 |
| Electron exposure (e$^-$/Å$^2$) | 48.0 |
| Defocus range (μm) | 1.2–3.0 |
| Pixel size (Å) | 0.822 |
| Symmetry imposed | C1 |
| Initial particle images (no.) | 2,240,373 |
| Final particle images (no.) | 647,468 |
| Map resolution (Å) | 2.6 |
| FSC threshold | 0.143 |
| Map resolution range (Å) | 2.5–3.5 |
| *Refinement* | |
| Initial model used (PDB code) | 6EZ8 |
| Model resolution (Å) | 2.6 |
| FSC threshold | 0.143 |
| Model resolution range (Å) | 2.5–3.5 |
| Map sharpening $B$ factor (Å$^2$) | −42.3 |
| Model composition | |
| Non-hydrogen atoms | 20,899 |
| Protein residues | 2669 |
| Ligands | 0 |
| $B$ factors (Å$^2$) | |
| Protein | 47.14 |
| Ligand | N/A |
| R.m.s. deviations | |
| Bond lengths (Å) | 0.006 |
| Bond angles (°) | 1.102 |
| Validation | |
| MolProbity score | 2.11 |
| Clashscore | 14.85 |
| Poor rotamers (%) | 0.56 |
| Ramachandran plot | |
| Favoured (%) | 93.41 |
| Allowed (%) | 6.29 |
| Disallowed (%) | 0.30 |

*N/A* not applicable.

located at their $C_\alpha$ positions and interacting via appropriate bonding, bending, torsion-angle, and non-bonding potential. A Gō-like model[71] was employed to maintain the structured, globular domains as quasi-rigid in the simulation. The model was built based on the experimental EM structure of the complex (PDB 6X9O). The EM structure of the complex is missing ~26% of the residues. We assume that the residues with known coordinates form a quasi-rigid part of the complex while the residues with missing coordinates are flexible.

For the flexible regions, we adopt a simple model in which adjacent amino acids beads are joined together into a polymer chain by means of virtual bond and angle interactions with a quadratic potential.

$$V_b = K_b (b - b_0)^2; \; V_\alpha = K_\alpha (\alpha - \alpha_0)^2$$

with the constants $K_b = 50$ kcal/mol and $K_\alpha = 1.75$ kcal/mol and the equilibrium values $b_0 = 3.8$ Å and $\alpha_0 = 112°$ for bonds and angles, respectively. The excluded volume between non-bonded beads was treated with pure repulsive potential

$$V_R = \varepsilon_R (\sigma_R / r_{ij})^{12}$$

where $r_{ij}$ is the inter-bead distance, $\sigma_R = 4$ Å, and $\varepsilon_R = 2.0$ kcal/mol.

We used experimentally observed cross-links to improve the sampling of the flexible regions of the model. Because of the structural flexibility, not all observed cross-links are compatible with one another, meaning that there is no one conformation of the complex that has a geometry such that all cross-links can be formed. Therefore, only a subset of compatible cross-links can be included in MD simulations as distance restraints. We implemented a procedure where during MD simulation is run with a set of randomly selected restraints for a period of time, then a new set of randomly selected restraints was generated. This was repeated ~2000 times along the MD trajectory. A harmonic potential normally used for distance restraints is not suitable for this procedure, since a new randomly selected distance restraint could be incompatible with the current conformation (the corresponding $C_\alpha$–$C_\alpha$ distance is large and causes very large forces), which will lead MD simulation to terminate. This problem can be avoided by using the sigmoidal function as a potential for a distance restraint, since at large distances, the sigmoidal potential produces forces close to zero. This method has previously been used to successfully model other dynamic complexes[72].

To account for the experimentally observed cross-links, we introduced in the force field a distance restraint term given by the following potential:

$$V_{XL}(t) = \sum_{p=1}^{N_c} \sum_{k=1}^{N_{XL}} \delta_{\xi_p(t)}^k V_l^k; \; V_l^k = K_{XL} / (1 + e^{-\beta(l_k(t) - l_0)})$$

The sum is over all cross-links, $N_{XL}$ is the number of cross-links; $N_c$ is the number of selected active cross-links; $l_k$ is the $C_\alpha$–$C_\alpha$ distance for residues involved in $k$th cross-link; $l_0 = 25$ Å is the upper bounds for PhoX cross-links; $\beta = 0.5$ is the slope of the sigmoidal function; $K_{XL} = 10$ kcal/mol is the force constant; $\delta_i^k$ is the Kronecker delta; and $\xi_p(t)$ is the random digital number selected from the interval $[1, N_{XL}]$. We chose to keep active a small number, $N_c = 5$ randomly selected restraints, numbers $\xi_p(t)$, that are updated every $\tau_{XL} = 0.5$ ns during the MD simulation.

**Fitting structural ensemble to SAXS data**. The goodness-of-fit of an ensemble of structural models of the complex to the SAXS data was evaluated by comparing an ensemble average profile, $I_{avrg}(q)$, with the experimental one $I_{exp}(q)$.

$$\chi_{SAXS} = \left[ \frac{1}{N_q} \sum_{i=1}^{N_q} \left[ \frac{I_{exp}(q_i) - \alpha \cdot I_{avrg}(q_i)}{\sigma(q_i)} \right]^2 \right]^{1/2}$$

where,

$$\alpha = \sum_{i=1}^{N_q} I_{exp}(q_i) \cdot I_{avrg}(q_i) / \sum_{i=1}^{N_q} I_{exp}(q_i) \cdot I_{exp}(q_i),$$

and,

$$I_{avrg}(q_i) = \sum_{k=1}^{N_{ens}} I_{calc}^k(q_i) \cdot w_k$$

Here $I_{calc}^k(q)$ is scattering intensity predicted for the $k$th conformation, $N_q$ is number of experimental points, $\sigma(q)$ is the experimental error, $N_{ens}$ is the number of conformations in the ensemble and $w_k$ is a weight of the $k$th conformation.

The optimal weights for each ensemble member were obtained with SES method[73] to minimise the discrepancy of the ensemble average profile from the experimental scattering data. Theoretical scattering profiles for each conformation in the ensemble were calculated in the $q$ range $0 < q < 0.30$ Å$^{-1}$ using FoXS[68].

**Validation of effects of polyglutamine expansion on ensemble structure with MD simulation**. In all, 800 ns long unconstrained MD trajectories were calculated for both HTT-HAP40 Q23 and HTT-HAP40 Q54 complexes, saving frames every 100 ps. We obtained two ensembles of structures, each consisting of 8000 models, that were used to analyse distances between lysine residues for which experimental cross-links were observed. Focussing on the cross-links that were observed for exon 1 and IDR residues (Supplementary Tables 3–6), we assessed the cross-link contact frequency of each cross-link in the ensemble.

**Size-exclusion chromatography multi-angle light scattering**. The absolute molar masses and mass distributions of purified protein samples of HTT-HAP40 Q23, HTT-HAP40 Q54 and HTT-HAP40 Δexon 1 at 1 mg/mL were determined using SEC-MALS. Samples were injected through a Superose 6 10/300 GL column (GE Healthcare) equilibrated in 20 mm HEPES, pH 7.5, 300 mm NaCl, 2.5% (v/v) glycerol and 1 mm TCEP followed in-line by a Dawn Heleos-II light scattering detector (Wyatt Technologies) and a 2414 refractive index detector (Waters). Molecular mass calculations were performed using ASTRA 6.1.1.17 (Wyatt Technologies) assuming a dn/dc value of 0.185 mL/g.

**In silico analysis of the HTT-HAP40 protein complex structure**. HTT-HAP40 models were analysed using Pymol[74] and APBS[75]. For conservation analysis, HTT and HAP40 orthologues were extracted from Ensembl, parsed to remove low quality or partial sequences and then aligned using Clustal[76]. Multiple sequence alignments were then analysed using Consurf[77] and conservation scores mapped to the HTT-HAP40 (PDBID: 6X9O) structure in Pymol. Ligand-able pocket analysis was completed as previously reported[78]. Briefly, HTT-HAP40 model PDB files were loaded in ICM (Molsoft, San Diego), optimal positions of added polar hydrogens were generated and correct orientation of side-chain amide groups for glutamine and asparagine and most favourable histidine isomers were identified. The PocketFinder algorithm implemented in ICM, which uses a transformation of the Lennard–Jones potential to identify ligand-binding envelopes regardless of the presence of bound ligands, was then applied[79]. Residues with side-chain heavy atoms within 2.8 Å of the molecular envelope were identified as lining the pocket.

**Statistics and reproducibility**. Experiments were performed at least 2–3 times in distinct technical replicates to confirm reproducibility. Sample sizes and statistical analyses used in this study are described above in "Methods" and are also detailed in figure legends.

**Reporting summary**. Further information on research design is available in the Nature Research Reporting Summary linked to this article.

## Data availability

All Supplementary Data files can be accessed via Zenodo[80]. Raw and preprocessed mass spectrometry data used in this study is deposited in Figshare with identifier 839[81] and PRIDE through accession PXD028313. Also available through these links are CSM tables that show the Scores and CSMs identified in our XLMS data sets as well as mass error. Cryo-EM maps can be downloaded at EMDB 22106 and model coordinates at PDBID 6X9O. All expression constructs are available through Addgene.

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

## Acknowledgements

We acknowledge the use of the SAXS Core Facility of the Center for Cancer Research (CCR), NCI, National Institutes of Health. NCI SAXS Core is funded by FNLCR contract HHSN261200800001E and the intramural research programme of the NIH, NCI, CCR. This research used 12-ID-B beamline of the Advanced Photon Source, a United States Department of Energy (DOE) Office of Science User Facility operated for the DOE Office of Science by Argonne National Laboratory under Contract No. DE-AC02-06CH11357. This research was supported by the CHDI Foundation (to R.J.H., C.H.A., J.B.C.), the Huntington Society of Canada (to R.J.H., C.H.A.), the Wellcome Trust #219477 (to S.M.L., J.D.) and the EU Horizon 2020 programme INFRAIA project Epic-XS Project 823839 (to J.F.H., S.T., A.J.R.H.). R.J.H. is the recipient of the Huntington's Disease Society of America Berman Topper Career Development Fellowship. The Structural Genomics Consortium is a registered charity (no: 1097737) that receives funds from AbbVie, Bayer AG, Boehringer Ingelheim, Genentech, Genome Canada through Ontario Genomics Institute [OGI-196], the EU and EFPIA through the Innovative Medicines Initiative 2 Joint Undertaking [EUbOPEN grant 875510], Janssen, Merck KGaA (aka EMD in Canada and US), Pfizer, Takeda and the Wellcome Trust [106169/ZZ14/Z].

## Author contributions

CryoEM experiments and data processing was completed by J.D.; all mass spectrometry experiments were completed by J.F.H. and S.T.; SAXS data collection was completed by X.Z.; mouse experiments were completed by J.P.C.; cell biology experiments were completed by N.B. and S.G.; modelling experiments were completed by A.L.; HTT/caspase-6 western blots were completed by M.M.S.; cloning and baculoviral production was completed by P.L., A.S. and A.H.; all other experiments were completed by R.J.H. R.J.H. conceived the project, designed and conducted experiments, analysed and interpreted data, supervised the project and wrote the manuscript. J.D., J.F.H., S.T., A.L., J.P.C., M.S., N.B. and X.Z. designed and conducted experiments, analysed and interpreted data and contributed to drafting and editing the manuscript. M.M.S., A.H., A.S., P.L. and S.G. conducted experiments and analysed data. R.T., A.J.R.H., J.B.C., C.H.A., S.M.L. and L.F. supervised the work, analysed and interpreted data and contributed to drafting and editing the manuscript.

## Competing interests

The authors declare no competing interests.
