## [Peer Review File · Communications Biology]

Reviewers' comments:

Reviewer #1 (Remarks to the Author):

This reviewer acknowledges that the authors put really great effort to generate integrative data and all data are technically sound. The major concern/issue of this manuscript is that the authors over-interpreted their data, which were pointed out in the previous comments by this reviewer (See the blow). These concerns still remain in the revised manuscript. For the publication, the data-interpretation should be more objective, as we can say only to the extent what the data actually show.

For examples,

1) Regarding the title "HAP40 orchestrates huntingtin structure for differential interaction with polyglutamine expanded exon 1", the authors mentioned in the rebuttal "The title of our manuscript reflects how HAP40 orchestrates the global structure of the HEAT domains of HTT which is the same in all forms of HTT." However, there is no data showing that HAP40 orchestrates the differential binding. It simply shows that there are additional bindings/crosslinks at exon1 of HTTQ57-HAP40 compared with HTTQ23-HAP40. To claim that HAP40 orchestrates the differential binding, the author should do the same analysis with HTTQ23 and HTTQ57 in the absence and presence of HAP40 and show that there is actual difference, which may not be possible due to self-oligomerization as the author mentioned. Therefore, the author should tone down and more accurately describe/interpret their data.

2) While the author set a subtitle "HTT and HAP40 protein levels are interdependent". The author did not show the change of HTT level in the absence of HAT40. All data show that the level of HAP40 is dependent on the level of HTT. Therefore, the author cannot say the level of HTT and HAP40 is interdependent.

3) The author claimed that the improved resolution of the model identified subtle but important structural features. However, what aspects of these subtle differences are important? Although the author mentioned PROTACs or PET ligands, there is little difference in the qualities of cryo-EM maps between this and previous works as the author illustrated in the response #2

4) Regarding XL-MS, I acknowledge that crosslinks of exon1 in HTTQ54-HAP40 has extra crosslinks compared with exon1 in HTTQ23-HAP40, but no differential crosslinks, as all crosslinks in Table4 (exon1 Q23) are found in Table 3 (exon1 Q23). As Fig. 7 was generated based on too little information, without other supporting experimental data, Figure 7 is still too speculative and can mislead what is actually happening in HTT exon 1.

These are reviewer comments on the rebuttals

Specifically, despite the high-resolution structure in this manuscript, this structure adds little information to the current information, which can be obtained from the previous structures (3.6 Å and 4.0Å, Guo et al. Nature 2018 and the recent structure) as the resolution of the previous structure is sufficient to map the interaction between HAP40 and HTT.

We thank the reviewer for their comments. Whilst we understand the reviewer's comments regarding the comparison with previous cryoEM models, we believe that the improved resolution of our model allows identification of subtle but important structural features of the HTT-HAP40 complex (Figure 2d and e). This improved resolution also enables small molecule discovery of this protein target, laying a foundation for the development of HTT ligands which might be adapted to PROTACs or PET ligands (see reply to

Reviewer #1 and
Figure 2h and Supplementary Table 2).

\RV: This reviewer acknowledge that the resolution of this study is better than previous works and there are extra helices visible in this structure. The author claimed that the improved resolution of the model identified subtle but important structural features. However, what aspects of this subtle difference are important? Although the author mentioned PROTACs or PET ligands, there are little difference in the qualities of cryo-EM maps between this and previous works as the author illustrated in the response #2

Furthermore, although the title says that HAP40 orchestrates huntingtin structure for differential interaction with polyQ exon1, only data showing the differential binding between Q23HTT-HAP40 and Q54HTT-HAP40 are based on several crosslinks between the exon 1 and other parts of HTT. These data do not tell anything about how HAP40 alters those interaction as there is no data comparing the differential XL-MS interactions between HTT and HTT-HAP40.

The title of our manuscript reflects how HAP40 orchestrates the global structure of the HEAT domains of HTT which is the same in all forms of HTT. The differential interactions observed upon polyglutamine expansion occur between this core HTT-HAP40 structure and the exon 1 region of HTT which are determined from analysis of 13 cross links for the Q23 form of the protein and 26 cross links for the Q54 form of the protein, now detailed in Supplementary tables 3 and 4 for additional clarity on the robustness of these observations. As we detail in the manuscript (lines 71-73) HTT in the absence of HAP40 is polydisperse, self-associates and cannot be reliably purified to yield just the monomer apo form of the protein (Harding et al (2019) JBC, Guo et al (2018) Nature). Therefore, it is extremely technically challenging to conduct reliable XL-MS studies of this sample without the risk of intermolecular cross-links which would obfuscate the dataset which is why we have instead pursued this analysis with the HTT-HAP40 series of constructs.

\RV: There is no data showing that HAP40 orchestrates the differential binding. It simply shows that there is additional bindings/crosslinks at exon1 of HTTQ57-HAP40 compared with HTTQ23-HAP40. To claim that HAP40 orchestrates the differential binding, the author should do the same analysis with HTTQ23 and HTTQ57 in the absence and presence of HAP40 and show that there is actual difference, which may not be possible due to self-oligomerization as the author mentioned. Therefore, the author should tone down and more accurately describe/interpret their data. Regarding XL-MS, I acknowledge that crosslinks of exon1 in HTTQ54-HAP40 has extra crosslinks compared with exon1 in HTTQ23-HAP40, but no differential crosslinks, as all crosslinks in Table4 (exon1 Q23) are found in Table 3 (exon1 Q23). As Fig. 7 was generated based on too little information, without other supporting experimental data, Figure 7 is still too speculative can mislead what is actually happening in HTT exon 1.

Lastly, the conclusions regarding the flexibility of N-terminal exon1 is not sufficiently supported by the presented data presented. The authors modeled the complete structures of HTT-HAP40 focusing on exon 1 based on cryo-EM, SAXS and XL-MS data. However, the skewed interaction of the exon-1 Q54 to Q23 is based on only several crosslinks observed between the exon-a and a.a.1300-1500 region without further supporting

data.

As mentioned above, in our respective datasets, we identify 13 cross links for exon 1 for the Q23 form of the protein and 26 cross links for exon 1 for the Q54 form of the protein, now detailed in Supplementary Figure 5c, Supplementary tables 3 and 4 for additional clarity regarding this data and our subsequent analysis of changes to exon 1 upon polyglutamine expansion. This clearly shows that exon 1 in the Q54 form of the protein makes many more contacts to the complex surface, spanning a greater number of regions of surface area. Additionally, we performed contact frequency analysis (see response to Rev #4 and new manuscript section, lines 386-399) which indicates that the difference in exon 1 conformation between Q54 and Q23 forms of HTT-HAP40 is significant. Thus, we believe this supports our conclusion that polyglutamine expansion modulates the conformational space occupied by HTT exon 1.
\RV: See the above.

Other major issue.

1. Fig. 1 shows that the abundance of HAP40 is correlated with the level of HTT. The level (0.3) of HAP40 protein in LKO is reduced to the half of the level (0.7) in WT. Does it mean that the half of HAP40 existing in cell forms a complex with HTT? Throughout the manuscript, the authors mentioned that HTT is stabilized upon HAP40 binding. However, Fig. 1 simply shows that HAP40 seems to be stabilized in the presence of HTT. In order to claim that the HTT is stabilized in the presence of HAP40, the authors should examine if the level of HTT is affected by the absence of HAP40. In addition, the authors should show that mRNA level of HAP40 is not affected in the absence of HTT.

We thank the reviewer for their suggestion and have updated the data shown in Figure 1a-d to include an analysis of both the protein and RNA levels of Htt and Hap40 in the wildtype (WT) and liver-knockout mouse (LKO). The analysis of the RNA levels indicates that there is no change in Hap40 RNA levels upon Htt knock out. We agree with the reviewer that this does suggest that Hap40 protein stability and/or abundance is dependent on HTT protein levels and have amended the text to reflect this (lines 126-128). However, our protein level analysis shows good correlation in the amount by which both Htt and Hap40 decrease upon knockout. Please note that the LKO mouse is a hepatocyte specific knock out of Htt so a maximum of 80% reduction is expected in total liver tissue which includes other cell types. Additionally, we treated RPE1 cells with the huntingtin lowering drug branaplam. HTT levels and HAP40 levels were both lowered by branaplam treatment in a dose-dependent manner and the lowering of both proteins was correlated and statistically significant. Overall, these data suggest that HAP40 protein stability and/or abundance is dependent on HTT protein levels.

\RV: The author did not show the change of HTT level in the absence of HAT40. All data show that

the level of HAP40 is dependent on the level of HTT. Therefore, the author cannot say the level of HTT and HAP40 is interdependent. The author should change the subtitle accordingly, for example, 'HAP40 protein level is depend on the level of HTT'.

2. Despite this manuscript presented the high-resolution structure, this high-resolution structure does not add much information to the current information, which can be obtained from the previous structure (3.6 Å and 4.0Å) as the resolution of the previous structure is sufficient to map the interaction between HAP40 and HTT. The only differences are two additional helices of C-HEAT (32 amino acid) present in the new structure and two N-terminal helices of HAP40 absent. The authors should show the cryo-EM map fitted with amino acids at several regions of HTT to examine if this 2.6Å resolution structure really improved the quality of the map compared with the previous structures.

To address the reviewer's concerns, we have generated some comparison map figures (blue map in each panel corresponds to our map EMD-22106 and purple corresponds to Guo et al EMD-3984). However, determining comparable contour levels is extremely difficult given that the properties of the two maps are different which means that the absolute contour value for one map will not match the second map. Therefore, we have normalized the contour manually based on backbone density. Nonetheless, our map has much more clearly defined side chains and the overall model to map fit is better in that respect, for example, Phe residues in i) and ii); Lys/Arg across all three panels.

\RV: The author claimed that the improved resolution of the model identified subtle but important structural features. However, what aspects of these subtle differences are important? Although the author mentioned PROTACs or PET ligands, there is little difference in the qualities of cryo-EM maps between this and previous works as the author illustrated in the response #2

3. Regarding XL-MS, how many unique crosslinks out of total cross-links were mapped to the exon1 and the IDR region for each sample? What are the unique crosslinks only found in Q23-HAP40, Q54-HAP40 or exon1d-HAP40?

All of the information regarding the cross-links we identified in this study can be found in the XL-MS supplementary data file accompanying this submission, the Figshare data deposit linked in the "Materials and Correspondence" section of the manuscript (<https://figshare.com/s/39b5b1a81838cd21ea92>) and we have now also included Supplementary Figure 4, Supplementary Figure 5 which shows the validation and reproducibility of the crosslinks we have identified. While there are many shared cross-links between different HTT-HAP40 samples (50 common to all three samples, plus 47 common to 2 samples), there are at least 30 unique cross-links for each sample which generally span the range of the protein structure. Regarding exon 1, there are 1 and 14 unique crosslinks for HTT-HAP40 Q23 and Q54 respectively. For the IDR, there are 4, 5 and 2 unique crosslinks for HTTHAP40Q23, Q54 or Δexon 1 respectively.

\RV: Seeing the Suppl Tables and Figures, it is more correct to say that HTTQ54-HAP40 has additional crosslinks/binding not differential binding/crosslinks as HTTQ54-HAP40 has almost all

crosslinks that HTTQ23-HAP40 has.

4. Based on the integrated model, the authors concluded that the polyQ expansion affect the conformational dynamics of the exon1 and the IDR. However, the constrains that the authors used for the integrative modeling are several XL-MS and SAXS envelop having subtle difference among Q23-HAP40, Q54-HAP40 or exon1d-HAP40. Therefore, the integrated model is highly speculative and not convincing. The similarity of the SAXS envelopes reflects the fact that HTT-HAP40 Q23, Q54 or Δ exon 1 complexes have the same ordered structural regions (75% of residues) which dominate the measurements. The flexible regions, that comprise $\sim 25\%$ of these complexes, are the most likely contributors to the significant observed differences in the SAXS envelopes. The fact that the model resolved by cryo-EM (which lacks the flexible regions) does not fit the SAXS intensity profile indicates that flexible regions of the HTT protein are contributing to the differential SAXS profiles compared to the cryo-EM model and relative to one another (as the only difference between the 3 SAXS samples are exon 1 lengths). There are 126 and 109 crosslinks for HTT-HAP40 Q23 and Q54, respectively, which is fairly large number of constraints to help model the conformations explored by the flexible part of the complexes. Our final models are in good agreement with all experimental data. In the absence of an experimentally determined full-length HTT structure, we believe our modelling provides important insight into the organisation of this protein.

\RV: Although the authors referred to 126 and 109 crosslinks, the model in Figure 7 focusing on exon1 is based on 13 and 26 crosslinks. I do acknowledge that there are 13-14 additional crosslinks in HTTQ53-HAP40 exon1 compared with HTTQ23-HAP40 exon1. Have the authors fitted the model shown in Figure7 to the SAXS envelope? Unless the models in Figure7 fit reasonably to the SAXS envelopes, the model in Figure 7 is too speculative and can be misleading, and should be supported by sufficient supporting experimental data.

Reviewer #2 (Remarks to the Author):

Harding et al. submit a revised version of the manuscript transferred from Nature Communications submission. They strengthen the claim of the evolutionary codependence of HTT and HAP40 by adding an experiment in human cells. They add a more convincing explanation of the suitability of their structure for drug development. They address some of the minor comments, which is acceptable.

My remaining concern is the appropriateness of using crosslinking to study disordered regions in general and the specific conclusions drawn. Again, I think this is the original part of the work, but in the revised version, I understand it even less than in the original ;-)

First, the authors respond about crosslink-induced aggregation and reproducibility, which had not been the point of my concern. By speculating: "I could imagine that in the case of disordered regions, crosslinking could distort the conformation through the covalent bonds. Or it could also bias the conformational equilibria by locking specific conformations. Perhaps it could also lead to non-native conformations induced by the accumulation of multiple crosslinks within the same protein molecule" I meant crosslinks WITHIN a single molecule, not crosslinks between molecules leading to aggregation.

The authors add two references but I am not sure about what do they mean, could it be accompanied with some comment on how crosslinks can be used to study disordered region?

Second, to address my concerns, the authors added unconstrained MD simulations, which indeed might be the way to address this. I do not, however, understand the conclusions from the results. If I understand correctly, the authors say that contact frequencies between lysine residues of exon-1 to the C-HEAT, derived from MD, are similar for both Q23 and Q54 isoforms. If so, it would be conflicting with the XL-MS data, which led to differential crosslinks between Q23 and Q54. But the authors say it is supporting the results. Perhaps this is a matter of more straightforward explanations in the text and the supplementary figure 7 (which is difficult to navigate, e.g., what does the green, blue, and patterned scheme mean? Is b just a subset of a? and what is b—in one place, it says, "Results are shown for 13 crosslinks that involve exon1 N-terminal residues," and in another place, line 147 of the supplementary figure file, it says it is IDR? And why in b there is no comparison to Q54? Aren't the graphs in c cropped at 11 %?).

Getting confused on the supplementary figure 7, I now have similar trouble with Supplementary Figure 6d. There, MD shows that Q23 could theoretically make some crosslinks experimentally observed exclusively for Q54. This observation was used as an argument that "the additional crosslinks observed for the polyglutamine expanded form of HTT-HAP40 may not be driven solely by the length of the exon 1 region". But, couldn't that also mean that the MD approach used does not agree with XL-MS data, invalidating the entire analysis in Supplementary Figure 7?

I am lost! Forgive me if this is just my impatience in reading and I am looking forward to clearer explanations.

Reviewer #3 (Remarks to the Author):

The authors have addressed all my comments sufficiently.

The only remaining aspect is the data upload to an official repository (e.g., PRIDE). This is usually required by all nature journals and is also strongly recommended by the MS community. Upload to figshare is not at all acceptable. In addition, the uploaded folder cannot be opened by windows.

Reviewers' comments:

Original review comments from reviewers

First round of response from authors

Second round of response from reviewers

Second round of response from authors

NB: changes to the manuscript and supplementary files in response to the reviewer's comments are highlighted in yellow as well as in track changes

Reviewer #1 (Remarks to the Author)

This reviewer acknowledges that the authors put really great effort to generate integrative data and all data are technically sound. The major concern/issue of this manuscript is that the authors over-interpreted their data, which were pointed out in the previous comments by this reviewer (See the blow). These concerns still remain in the revised manuscript. For the publication, the data-interpretation should be more objective, as we can say only to the extent what the data actually show.

For examples,

1) Regarding the title "HAP40 orchestrates huntingtin structure for differential interaction with polyglutamine expanded exon 1", the authors mentioned in the rebuttal "The title of our manuscript reflects how HAP40 orchestrates the global structure of the HEAT domains of HTT which is the same in all forms of HTT." However, there is no data showing that HAP40 orchestrates the differential binding. It simply shows that there is additional bindings/crosslinks at exon1 of HTTQ57-HAP40 compared with HTTQ23-HAP40. To claim that HAP40 orchestrates the differential binding, the author should do the same analysis with HTTQ23 and HTTQ57 in the absence and presence of HAP40 and show that there is actual difference, which may not be possible due to self-oligomerization as the author mentioned. Therefore, the author should tone down and more accurately describe/interpret their data.

The original title of this manuscript describes how HAP40 orchestrates the global structure of the HTT molecule, organising its different HEAT domains into the stable structure we observe by cryoEM. This aspect of the structural organisation of HTT is independent of the polyglutamine expansion, as we have described within this paper and as has been described in the published literature (Harding *et al* (2019) JBC, Huang *et al* (2021) Structure). The globular structure of HTT, orchestrated through the binding of HTT to HAP40 and resolved in our cryoEM data, subsequently has differential interactions with the flexible exon 1 region of HTT, depending on the polyglutamine expansion of this region which we observe in our cross-linking studies. That said, we have taken the reviewers comments on board and resubmit this manuscript under an amended title "Huntingtin structure is orchestrated by HAP40 and shows a polyglutamine expansion-specific interaction with exon 1" which we believe more precisely describes two of the key findings of the paper, firstly that through interaction with HAP40, HTT is organised into a highly stable complex, and secondly, that the flexible exon 1 region interacts with the globular structure of HTT and that there are additional exon 1/HTT interactions in the presence of polyglutamine expansion. Experiments in the absence of HAP40 will almost certainly be confounded by HTT self-association, as the reviewer suggests. We believe that our new title is semantically accurate for the data presented. However, if the editor disagrees with our assertion, we could put forward some alternatives.

2) While the author set a subtitle 'HTT and HAP40 protein levels are interdependent'. The author did not show the change of HTT level in the absence of HAT40. All data show that the level of HAP40 is dependent on the level of HTT. Therefore, the author cannot say the level of HTT and HAP40 is interdependent.

We thank the reviewer for this comment and agree that this relationship could be described more accurately. Therefore, we have amended the language to "HAP40 levels are dependent on HTT" (see line 106), "...thereby contributing to a richer understanding of HTT and its relationship with HAP40." (lines 442-443) and "Overall, this suggests an HAP40 protein stability and/or abundance is dependent on HTT protein levels" (see lines 453-454).

3) The author claimed that the improved resolution of the model identified subtle but important structural features. However, what aspects of these subtle differences are important? Although the author mentioned PROTACs or PET ligands, there is little difference in the qualities of cryo-EM maps between this and previous works as the author illustrated in the response #2

Our high resolution cryoEM structure of HTT-HAP40 provides detailed insight into the precise organisation of both HTT and HAP40 as well as the interface between these two proteins. Additionally, our structure largely validates the modelling of other lower resolution structures (6EZ8 – 4 Å, 7DXJ – 3.6 Å, 7DXH – 4.1 Å) which were determined with poorer quality maps. As **Figure 2d** illustrates, our higher resolution structure resolves additional helices and also shows a relative shift of the C-HEAT domain in the organisation of the complex relative to the lower resolution structures published to date referenced above. Given that HAP40 is the only experimentally and structurally validated interaction partner of HTT, we argue that detailed knowledge of the interaction interface and complex structure is important to understand this complex and HTT structure-function.

As per our previous rebuttal and the supplied figure (**Figure R1** in this document), our higher resolution map allows confident positioning of side chains and other structural features of the HTT-HAP40 complex. Additionally, as we drew attention to the reviewer in our previous rebuttal, the improvement in the previously published map and our map can be clearly seen in the local

resolution maps of the different datasets i.e. **Supplementary Figure 1b** in our manuscript and Extended Data Figure 2d of Guo et al (2018) and Figure S1 of Huang et al (2021). This clearly shows that the overall resolution of our structure is improved compared to all other available datasets.

The implications of our higher resolution for future drug discovery are significant. As is well described in the literature, structure-based drug discovery (SBDD) hinges on the availability of high-resolution structural models, frequently quoted as below a critical threshold of 2.5-3.0 Å resolution. High resolution structures which fall below this threshold, such as our 6X90 model, allow more accurate placement of amino acid side chains and therefore a greater reliability and accuracy of the resultant surface 'pockets' into which drug-like small molecules can bind. Thus, calculations of potential druggability, computational ligand docking into pockets/surfaces as well as virtual ligand screening (VLS) and AI approaches for computational prediction of ligand binding are all much more likely to be successful with our structure than for the previous structures.

Please see references regarding this topic:

- Subramaniam et al (2016) *Curr Opin Struct Biol* <https://dx.doi.org/10.1016%2Fj.sbi.2016.07.009>
- Anderson et al (2003) *Chem Biol* <https://doi.org/10.1016/j.chembiol.2003.09.002>
- Rawson et al (2018) *PNAS* <https://dx.doi.org/10.1073%2Fpnas.1708839115>
- Nakane et al (2020) *Nature* <https://doi.org/10.1038/s41586-020-2829-0>
- Danev et al (2021) *Nat Comms* <https://doi.org/10.1038/s41467-021-24650-3>
- Zhang et al (2021) *Structure* <https://doi.org/10.1016/j.str.2021.04.008>

Therefore, we believe that our comments in the manuscript regarding the improvement in resolution are not only based entirely in fact but are highly relevant for future efforts for SBDD to develop HTT ligands.

Figure R1 - Comparison map figures. Blue map in each panel corresponds to our map EMD-22106 and purple corresponds to Guo et al EMD-3984. We have normalized the contour across the two maps manually based on backbone density. Nonetheless, our map has much more clearly defined side chains and the overall model-to-map fit is better in that respect, for example, Phe residues in i) and ii); Lys/Arg across all three panels. Examples of such residues with better defined maps and more accurate conformations are pointed out with arrows in each panel.

4) Regarding XL-MS, I acknowledge that crosslinks of exon1 in HTTQ54-HAP40 has extra crosslinks compared with exon1 in HTTQ23-HAP40, but no differential crosslinks, as all crosslinks in Table4 (exon1 Q23) are found in Table 3 (exon1 Q23). As Fig. 7 was generated based on too little information, without other supporting experimental data, Figure 7 is still too speculative can mislead what is actually happening in HTT exon 1.

The cross-links made by exon1 in the Q23 form of HTT-HAP40 compared to the Q54 form are significantly non-identical, albeit overlapping. Therefore, we believe 'differential' is an accurate term as the cross-linking lists are non-identical/different.

Regarding our conformational models in Figure 7 which are based on three experimental methods (Cryo-EM, SAXS and crosslinking) plus molecular dynamic simulations: We would argue that numerous structural models, such as we have generated for this manuscript, have been published on the basis of similar integrative structural approaches, and that our model is not misleading but should be treated with the appropriate degree of caution any informed reader should consider such structural models with.

Please see published examples of other structural models, generated on the basis of comparable data:

- Guitoli et al (2016) *PNAS* <https://dx.doi.org/10.1073%2Fpnas.1523708113> (generation of a full-length structural model of LRRK2 using EM, cross-linking and SAXS)
- Klykov et al (2018) *PNAS* <https://dx.doi.org/10.1073%2Fpnas.1911785117> (integrative structural approach using cross-linking to build models of fibrinogen oligomers)

- Kim et al (2018) Nature <https://doi.org/10.1038/nature26003> (integrative structural approach to generate a model of the nuclear pore complex using low resolution cryoEM data and cross-linking)
- Harwood et al (2021) Mol Cell Proteomics <https://doi.org/10.1016/j.mcpro.2021.100090> (integrated negative stain electron microscopy (EM), small-angle X-ray scattering (SAXS), and cross-linking–mass spectrometry (XL-MS) to model native A2M)

Additionally, please see these references regarding application of cross-linking to understand disordered protein structures here:

- McDonald et al (2019) PNAS <https://doi.org/10.1016/j.str.2019.03.008> (cross-linking used to model different conformation of PrP),
- Arlt et al (2016) Ange Chem <https://doi.org/10.1002/anie.201609826> (cross-linking analysis of p53 IDRs),
- Chen et al (2019) Nat Comm <https://doi.org/10.1038/s41467-019-10355-1> (cross-linking used to model conformational changes of the IDP tau).

We believe that these examples show that our approach is not only appropriate but a recognised and supported approach for understanding challenging protein structures within the structural biology community.

These are reviewer comments on the rebuttals

Specifically, despite the high-resolution structure in this manuscript, this structure adds little information to the current information, which can be obtained from the previous structures (3.6 Å and 4.0Å, Guo et al. Nature 2018 and the recent structure) as the resolution of the previous structure is sufficient to map the interaction between HAP40 and HTT.

We thank the reviewer for their comments. Whilst we understand the reviewer's comments regarding the comparison with previous cryoEM models, we believe that the improved resolution of our model allows identification of subtle but important structural features of the HTT-HAP40 complex (Figure 2d and e). This improved resolution also enables small molecule discovery of this protein target, laying a foundation for the development of HTT ligands which might be adapted to PROTACs or PET ligands (see reply to Reviewer #1 and Figure 2h and Supplementary Table 2).

RV: This reviewer acknowledge that the resolution of this study is better than previous works and there are extra helices visible in this structure. The author claimed that the improved resolution of the model identified subtle but important structural features. However, what aspects of this subtle difference are important? Although the author mentioned PROTACs or PET ligands, there are little difference in the qualities of cryo-EM maps between this and previous works as the author illustrated in the response #2

Please see response above regarding the resolution of our model, the differences and details this highlights and the subsequent opportunity for downstream applications such as structure-based drug discovery.

Furthermore, although the title says that HAP40 orchestrates huntingtin structure for differential interaction with polyQ exon1, only data showing the differential binding between Q23HTT-HAP40 and Q54HTT-HAP40 are based on several crosslinks between the exon 1 and other parts of HTT. These data do not tell anything about how HAP40 alters those interaction as there is no data comparing the differential XL-MS interactions between HTT and HTT-HAP40.

The title of our manuscript reflects how HAP40 orchestrates the global structure of the HEAT domains of HTT which is the same in all forms of HTT. The differential interactions observed upon polyglutamine expansion occur between this core HTT-HAP40 structure and the exon 1 region of HTT which are determined from analysis of 13 cross links for the Q23 form of the protein and 26 cross links for the Q54 form of the protein, now detailed in Supplementary tables 3 and 4 for additional clarity on the robustness of these observations. As we detail in the manuscript (lines 71-73) HTT in the absence of HAP40 is polydisperse, self-associates and cannot be reliably purified to yield just the monomer apo form of the protein (Harding et al (2019) JBC, Guo et al (2018) Nature). Therefore, it is extremely technically challenging to conduct reliable XL-MS studies of this sample without the risk of intermolecular cross-links which would obfuscate the dataset which is why we have instead pursued this analysis with the HTT-HAP40 series of constructs.

RV: There is no data showing that HAP40 orchestrates the differential binding. It simply shows that there is additional bindings/crosslinks at exon1 of HTTQ57-HAP40 compared with HTTQ23-HAP40. To claim that HAP40 orchestrates the differential binding, the author should do the same analysis with HTTQ23 and HTTQ57 in the absence and presence of HAP40 and show that there is actual difference, which may not be possible due to self-oligomerization as the author mentioned. Therefore, the author should tone down and more accurately describe/interpret their data. Regarding XL-MS, I acknowledge that crosslinks of exon1 in HTTQ54-HAP40 has extra crosslinks compared with exon1 in HTTQ23-HAP40, but no differential crosslinks, as all crosslinks in Table4 (exon1 Q23) are found in Table 3 (exon1 Q23). As Fig. 7 was generated based on too little information, without other supporting experimental data, Figure 7 is still too speculative can mislead what is actually happening in HTT exon 1.

Please see our comments above regarding an explanation for the title for this manuscript. We would argue that the evidence is quite clear that HAP40 orchestrates HTT structure as can be seen in our cryoEM datasets for apo and HAP40-bound HTT (Figures 3 and 2 respectively). As we explained previously, comparable cross-linking studies or other biophysical analyses with apo HTT samples are unlikely to yield meaningful data due to the issues with self-association. Please see above comments which detail how our integrative approach and model building is appropriate given similar approaches to tackle challenging structures which have been published in highly-respected journals. We do however agree that we can moderate the language in the description of these models and have made the following changes:

Line 368 – “As expected from our cross-linking results, **the conformations** adopted by exon 1 in the ensemble model of Q54 HTT-HAP40 complex are skewed compared to the Q23 ensemble...” to “As expected from our cross-linking results, **the suggested conformations** adopted by exon 1 in the ensemble model of Q54 HTT-HAP40 complex are skewed compared to the Q23 ensemble...”

Line 374 – “Exon 1 of our HTT-HAP40 **Q54 ensemble explores** a larger volume of conformational space...” to “Exon 1 of our HTT-HAP40 **Q54 ensemble appears to explore** a larger volume of conformational space...”

Line 386 – “For the HTT-HAP40 Q54 model **ensemble where exon 1** adopts the most diverse conformations...” to “For the HTT-HAP40 Q54 model **ensemble which suggests exon 1** adopts the most diverse conformations”

Line 484 – “**We demonstrate clear and novel structural differences** between the unexpanded and expanded forms of exon 1..” to “**Our data indicate potential structural differences** between the unexpanded and expanded forms of exon 1..”

Lastly, the conclusions regarding the flexibility of N-terminal exon1 is not sufficiently supported by the presented data presented. The authors modeled the complete structures of HTT-HAP40 focusing on exon 1 based on cryo-EM, SAXS and XL-MS data. However, the skewed interaction of the exon-1 Q54 to Q23 is based on only several crosslinks observed between the exon-a and a.a.1300-1500 region without further supporting data.

As mentioned above, in our respective datasets, we identify 13 cross links for exon 1 for the Q23 form of the protein and 26 cross links for exon 1 for the Q54 form of the protein, now detailed in Supplementary Figure 5c, Supplementary tables 3 and 4 for additional clarity regarding this data and our subsequent analysis of changes to exon 1 upon polyglutamine expansion. This clearly shows that exon 1 in the Q54 form of the protein makes many more contacts to the complex surface, spanning a greater number of regions of surface area. Additionally, we performed contact frequency analysis (see response to Rev #4 and new manuscript section, lines 386-399) which indicates that the difference in exon 1 conformation between Q54 and Q23 forms of HTT-HAP40 is significant. Thus, we believe this supports our conclusion that polyglutamine expansion modulates the conformational space occupied by HTT exon 1.

RV: See the above.

Other major issue.

1. Fig. 1 shows that the abundance of HAP40 is correlated with the level of HTT. The level (0.3) of HAP40 protein in LKO is reduced to the half of the level (0.7) in WT. Does it mean that the half of HAP40 existing in cell forms a complex with HTT? Throughout the manuscript, the authors mentioned that HTT is stabilized upon HAP40 binding. However, Fig. 1 simply shows that HAP40 seems to be stabilized in the presence of HTT. In order to claim that the HTT is stabilized in the presence of HAP40, the authors should examine if the level of HTT is affected by the absence of HAP40. In addition, the authors should show that mRNA level of HAP40 is not affected in the absence of HTT.

We thank the reviewer for their suggestion and have updated the data shown in Figure 1a-d to include an analysis of both the protein and RNA levels of Htt and Hap40 in the wildtype (WT) and liver-knockout mouse (LKO). The analysis of the RNA levels indicates that there is no change in Hap40 RNA levels upon Htt knock out. We agree with the reviewer that this does suggest that Hap40 protein stability and/or abundance is dependent on HTT protein levels and have amended the text to reflect this (lines 126-128). However, our protein level analysis shows good correlation in the amount by which both Htt and Hap40 decrease upon knockout. Please note that the LKO mouse is a hepatocyte specific knock out of Htt so a maximum of 80% reduction is expected in total liver tissue which includes other cell types. Additionally, we treated RPE1 cells with the huntingtin lowering drug branaplam. HTT levels and HAP40 levels were both lowered by branaplam treatment in a dose-dependent manner and the lowering of both proteins was correlated and statistically significant. Overall, these data suggest that HAP40 protein stability and/or abundance is dependent on HTT protein levels.

RV: The author did not show the change of HTT level in the absence of HAT40. All data show that the level of HAP40 is dependent on the level of HTT. Therefore, the author cannot say the level of HTT and HAP40 is interdependent. The author should change the subtitle accordingly, for example, ‘HAP40 protein level is depend on the level of HTT’.

Please see response above regarding the edits to the language we have made regarding the relationship between HTT and HAP40 protein levels.

2. Despite this manuscript presented the high-resolution structure, this high-resolution structure does not add much information to the current information, which can be obtained from the previous structure (3.6 Å and 4.0Å) as the resolution of the previous structure is sufficient to map the interaction between HAP40 and HTT. The only differences are two additional helices of C-HEAT (32 amino acid) present in the new structure and two N-terminal helices of HAP40 absent. The authors should show the cryo-EM map fitted with amino acids at several regions of HTT to examine if this 2.6Å resolution structure really improved the quality of the map compared with the previous structures.

To address the reviewer's concerns, we have generated some comparison map figures (blue map in each panel corresponds to our map EMD-22106 and purple corresponds to Guo et al EMD-3984). However, determining comparable contour levels is extremely difficult given that the properties of the two maps are different which means that the absolute contour value for one map will not match the second map. Therefore, we have normalized the contour manually based on backbone density. Nonetheless, our map has much more clearly defined side chains and the overall model to map fit is better in that respect, for example, Phe residues in i) and ii); Lys/Arg across all three panels.

RV: The author claimed that the improved resolution of the model identified subtle but important structural features. However, what aspects of these subtle differences are important? Although the author mentioned PROTACs or PET ligands, there is little difference in the qualities of cryo-EM maps between this and previous works as the author illustrated in the response #2

Please see response above regarding the resolution of our model, differences and details this highlights, and downstream applications for structure based drug discovery.

3. Regarding XL-MS, how many unique crosslinks out of total cross-links were mapped to the exon1 and the IDR region for each sample? What are the unique crosslinks only found in Q23-HAP40, Q54-HAP40 or exon1d-HAP40?

All of the information regarding the cross-links we identified in this study can be found in the XL-MS supplementary data file accompanying this submission, the Figshare data deposit linked in the "Materials and Correspondence" section of the manuscript (<https://figshare.com/s/39b5b1a81838cd21ea92>) and we have now also included Supplementary Figure 4, Supplementary Figure 5 which shows the validation and reproducibility of the crosslinks we have identified. While there are many shared cross-links between different HTT-HAP40 samples (50 common to all three samples, plus 47 common to 2 samples), there are at least 30 unique cross-links for each sample which generally span the range of the protein structure. Regarding exon 1, there are 1 and 14 unique crosslinks for HTT-HAP40 Q23 and Q54 respectively. For the IDR, there are 4, 5 and 2 unique crosslinks for HTTHAP40Q23, Q54 or Δexon 1 respectively.

RV: Seeing the Suppl Tables and Figures, it is more correct to say that HTTQ54-HAP40 has additional crosslinks/binding not differential binding/crosslinks as HTTQ54-HAP40 has almost all crosslinks that HTTQ23-HAP40 has.

The unique cross-links in the dataset for HTT-HAP40 Q54 are described as additional in the manuscript. However, we argue that as the contacts made by exon1 in our Q23 and Q54 datasets are non-identical, albeit overlapping and therefore describing them as differential contacts/interactions is appropriate.

4. Based on the integrated model, the authors concluded that the polyQ expansion affect the conformational dynamics of the exon1 and the IDR. However, the constrains that the authors used for the integrative modeling are several XL-MS and SAXS envelop having subtle difference among Q23-HAP40, Q54-HAP40 or exon1d-HAP40. Therefore, the integrated model is highly speculative and not convincing.

The similarity of the SAXS envelopes reflects the fact that HTT-HAP40 Q23, Q54 or Δexon 1 complexes have the same ordered structural regions (75% of residues) which dominate the measurements. The flexible regions, that comprise ~ 25% of these complexes, are the most likely contributors to the significant observed differences in the SAXS envelopes. The fact that the model resolved by cryo-EM (which lacks the flexible regions) does not fit the SAXS intensity profile indicates that flexible regions of the HTT protein are contributing to the differential SAXS profiles compared to the cryo-EM model and relative to one another (as the only difference between the 3 SAXS samples are exon 1 lengths). There are 126 and 109 crosslinks for HTT-HAP40 Q23 and Q54, respectively, which is fairly large number of constraints to help model the conformations explored by the flexible part of the complexes. Our final models are in good agreement with all experimental data. In the absence of an experimentally determined full-length HTT structure, we believe our modelling provides important insight into the organisation of this protein.

RV: Although the authors referred to 126 and 109 crosslinks, the model in Figure 7 focusing on exon1 is based on 13 and 26 crosslinks. I do acknowledge that there are 13-14 additional crosslinks in HTTQ53-HAP40 exon1 compared with HTTQ23-HAP40 exon1. Have the authors fitted the model shown in Figure7 to the SAXS envelope?

Unless the models in Figure 7 fit reasonably to the SAXS envelopes, the model in Figure 7 is too speculative and can be misleading, and should be supported by sufficient supporting experimental data.

We refer the reviewer to our previous comments addressing most of these concerns. As per our results and materials and methods sections (starting at lines 361 and 908 respectively), the integrative modelling approach does use the SAXS data as a constraint. The optimal ensembles shown in **Figure 7** were obtained by fitting to both SAXS scattering curve and all experimentally observed crosslinks.

Reviewer #2 (Remarks to the Author)

Harding et al. submit a revised version of the manuscript transferred from Nature Communications submission. They strengthen the claim of the evolutionary codependence of HTT and HAP40 by adding an experiment in human cells. They add a more convincing explanation of the suitability of their structure for drug development. They address some of the minor comments, which is acceptable.

My remaining concern is the appropriateness of using crosslinking to study disordered regions in general and the specific conclusions drawn. Again, I think this is the original part of the work, but in the revised version, I understand it even less than in the original ;-)

First, the authors respond about crosslink-induced aggregation and reproducibility, which had not been the point of my concern. By speculating: "I could imagine that in the case of disordered regions, crosslinking could distort the conformation through the covalent bonds. Or it could also bias the conformational equilibria by locking specific conformations. Perhaps it could also lead to non-native conformations induced by the accumulation of multiple crosslinks within the same protein molecule" I meant crosslinks WITHIN a single molecule, not crosslinks between molecules leading to aggregation. The authors add two references but I am not sure about what do they mean, could it be accompanied with some comment on how crosslinks can be used to study disordered region? The authors add two references but I am not sure about what do they mean, could it be accompanied with some comment on how crosslinks can be used to study disordered region?

Whilst it is a possibility that the overall structure of the protein might be contorted over time with an increasing number of crosslinks, we don't believe that this is a major concern under the conditions we used in these experiments which very lightly cross-link solvent accessible and proximal residues. If such an event were to occur, we would likely see decreased reproducibility of cross-links between replicate datasets whereas we see very consistent cross-linking when analysing replicate samples and also comparing the distinct Q23, Q54 and Δ exon 1 datasets (see **Supplementary Figures 4 and 5**).

Using cross-linking to study flexible or disordered regions of protein structure is well documented in the published literature. In addition to the references in the manuscript, please see these references regarding application of cross-linking to understand disordered protein structures here:

- Mitra et al (2020) Proteomics <https://doi.org/10.1002/pmic.202000011> (a review on the subject)
- McDonald et al (2019) PNAS <https://doi.org/10.1016/j.str.2019.03.008> (cross-linking used to model different conformation of PrP),
- Arit et al (2016) Ange Chem <https://doi.org/10.1002/anie.201609826> (cross-linking analysis of p53 IDRs)
- Chen et al (2019) Nat Comm <https://doi.org/10.1038/s41467-019-10355-1> (cross-linking used to model conformational changes of the IDP tau)

We believe that these examples show that our approach is not only appropriate but a recognised and supported approach for understanding challenging protein structures within the structural biology community.

We have amended the section highlighted by the reviewer (lines 361-363) so that it now reads: "Next, we modelled the complete structures of HTT-HAP40, including flexible and disordered regions, integrating our cryo-EM, SAXS and XL-MS data, similar to other studies using integrated approaches to study disordered protein structures^{38,39}."

Second, to address my concerns, the authors added unconstrained MD simulations, which indeed might be the way to address this. I do not, however, understand the conclusions from the results. If I understand correctly, the authors say that contact frequencies between lysine residues of exon-1 to the C-HEAT, derived from MD, are similar for both Q23 and Q54 isoforms. If so, it would be conflicting with the XL-MS data, which led to differential crosslinks between Q23 and Q54. But the authors say it is supporting the results. Perhaps this is a matter of more straightforward explanations in the text and the supplementary figure 7 (which is difficult to navigate, e.g., what does the green, blue, and patterned scheme mean? Is b just a subset of a? and what is b—in one place, it says, "Results are shown for 13 crosslinks that involve exon1 N-terminal residues," and in another place, line 147 of the supplementary figure file, it says it is IDR? And why in b there is no comparison to Q54? Aren't the graphs in c cropped at 11 %?).

Getting confused on the supplementary figure 7, I now have similar trouble with Supplementary Figure 6d. There, MD shows that Q23 could theoretically make some crosslinks experimentally observed exclusively for Q54. This observation was used as an argument that "the additional crosslinks observed for the polyglutamine expanded form of HTT-HAP40 may not be driven solely by the length of the exon 1 region". But, couldn't that also mean that the MD approach used does not agree with XL-MS data, invalidating the entire analysis in Supplementary Figure 7?

As the reviewer states, the contact frequencies for exon1-C-HEAT contacts are similar in our unconstrained simulations for HTT-HAP40 Q23 and Q54, whereas our experimental data shows a greater frequency of cross-links between exon1-C-HEAT in the Q54 sample. However, these are two very different analyses with different inputs and variables informing the simulations so a difference in outcome does not invalidate the approach in our opinion. The unconstrained modelling used in the contact frequency analysis would predict that there would be no difference when the movement of the flexible parts is unconstrained by the cross-linking information and is driven only by excluded volume and entropic forces in the calculations. However, our experimental data indicates this is not the case, supporting our hypothesis that there are likely additional biophysical properties of the exon1 region upon polyglutamine expansion which contribute to this disparity. We argue that this finding further supports the use of experimental constraints, such as the cross-linking data, to better inform our simulations.

To help readers better navigate this section we have amended the manuscript as follows:

- Supplementary Figure 6d has been removed so that this point may be more clearly addressed by the data in Supplementary Figure 7.
- Supplementary Figure 7 has been updated and annotated to more clearly guide the reader through the different panels. We have removed panel b which was superfluous to the thrust of our argument.
- Line 393 - The sentence: "All cross-links identified for the exon1 region of HTT-HAP40 Q23 are also observed for the Q54 form of the protein." to "All cross-links **experimentally** identified for the exon1 region of HTT-HAP40 Q23 are also observed for the Q54 form of the protein."
- Line 395 - The sentence: "However, exon 1-C-HEAT cross-links that are uniquely identified in our HTT-HAP40 Q54 experiments have similar frequencies in both simulations, supporting our conclusion that the experimental identification of exon 1-C-HEAT cross-links for HTT-HAP40 Q54 is significant (Supplementary Figure 7a)." has been edited to "However, exon 1-C-HEAT cross-links that are uniquely identified in our HTT-HAP40 Q54 experiments have **similar contact frequencies** in our unconstrained simulations of HTT-HAP40 Q23 and Q54, supporting our conclusion that the experimental identification of exon 1-C-HEAT cross-links for HTT-HAP40 Q54 is significant (Supplementary Figure 7a)."

The updated version of Supplementary Figure 7 is shown below:

Supplementary Figure 7. Validation of polyglutamine dependent structural changes to HTT-HAP40.

a Statistics of cross-links experimentally observed for HTT-HAP40 Q54 complex (numbered as per **Supplementary Table 2**) in two molecular dynamics ensembles calculated for HTT-HAP40 Q54 (**i, ii**) and HTT-HAP40 Q23 (**iii, iv**) complexes, respectively. Results for 26 cross-links that involve exon 1 N-terminal residues (K6 and K9) are shown. **i** and **iii** A cross-link Ca-Ca distance minimal in the ensemble consisting of 8,000 models obtained by unconstrained MD simulations. We assume that a cross-link could be formed when the corresponding Ca-Ca distance is $< 35 \text{ \AA}$. **ii** and **iv** Percentage of the models in the ensemble that have two Lys residues close enough to form a cross-link. **b** Statistics of cross-links experimentally observed for HTT-HAP40 Q23 (**i**) and HTT-HAP40 Q54 (**ii**) complexes that involve IDR residues of HTT (numbered as per **Supplementary Tables 4** and **5**). Percentage of the models in MD ensemble that have two Lys residues close enough to form a cross-link are shown for the ensembles of HTT-HAP40 Q23 (**i**) and HTT-HAP40 Q54 (**ii**) that were obtained by unconstrained MD simulations.

I am lost! Forgive me if this is just my impatience in reading and I am looking forward to clearer explanations.

Reviewer #3 (Remarks to the Author)

The authors have addressed all my comments sufficiently.

The only remaining aspect is the data upload to an official repository (e.g., PRIDE). This is usually required by all nature journals and is also strongly recommended by the MS community. Upload to figshare is not at all acceptable. In addition, the uploaded folder cannot be opened by windows.

We have uploaded the data to PRIDE as per the reviewer's request. This may be accessed through accession PXD028313 and the reviewer(s) may find the files through this login: Username: reviewer_pxd028313@ebi.ac.uk Password: 9I14MPfu. The data availability statement (line 1005) has been updated to "Raw and preprocessed mass spectrometry data used in this study is deposited in Figshare with identifier 839: (<https://figshare.com/s/39b5b1a81838cd21ea92>) and PRIDE through accession PXD028313."

REVIEWERS' COMMENTS:

Reviewer #1 (Remarks to the Author):

The revised manuscript reasonably addressed most of the concerns raised by this reviewer.

Reviewer #2 (Remarks to the Author):

My comments have been addressed to an acceptable extent.